# On the Unreasonable Effectiveness of Federated Averaging with Heterogeneous Data

**Jianyu Wang**[*]                                                    *jianyu.wang.thu@gmail.com*
*Carnegie Mellon University*

**Rudrajit Das**                                                          *rdas@utexas.edu*
*University of Texas at Austin*

**Gauri Joshi**                                                      *gaurij@andrew.cmu.edu*
*Carnegie Mellon University*

**Satyen Kale**                                                     *satyen@satyenkale.com*
*Google Research*

**Zheng Xu**                                                          *xuzheng@google.com*
*Google Research*

**Tong Zhang**                                                 *tongzhang@tongzhang-ml.org*
*University of Illinois Urbana-Champaign*

**Reviewed on OpenReview:** *https://openreview.net/forum?id=zF76Ga4EPs*

## Abstract

Existing theoretical results (such as (Woodworth et al., 2020a)) predict that the performance of federated averaging (FedAvg) is exacerbated by high data heterogeneity. However, in practice, FedAvg converges pretty well on several naturally heterogeneous datasets. In order to explain this seemingly unreasonable effectiveness of FedAvg that contradicts previous theoretical predictions, this paper introduces the *client consensus hypothesis*: the average of local models updates on clients starting from the optimum is close to zero. We prove that under this hypothesis, data heterogeneity does not exacerbate the convergence of FedAvg. Moreover, we show that this hypothesis holds for a linear regression problem and some naturally heterogeneous datasets such as FEMNIST and StackOverflow. Therefore, we believe that this hypothesis can better explain the performance of FedAvg in practice.

## 1 Introduction

Federated learning (FL) is an emerging distributed training paradigm (Kairouz et al., 2019; Wang et al., 2021), which enables a large number of clients to collaboratively train a powerful machine learning model without the need of uploading any raw training data. One of the most popular FL algorithms is Federated Averaging (FEDAVG), proposed by McMahan et al. (2017). In each round of FEDAVG, a small subset of clients are randomly selected to perform local model training. Then, the local model changes from clients are aggregated at the server to update the global model. This general local-update framework only requires infrequent communication between server and clients, and thus, is especially suitable for FL settings where the communication cost is a major bottleneck.

Due to its simplicity and empirical effectiveness, FEDAVG has become the basis of almost all subsequent FL optimization algorithms. Nonetheless, its convergence behavior – especially when the clients have

---

[*]Currently at Apple.

heterogeneous data – has not been fully understood yet. Existing theoretical results such as Woodworth et al. (2020a); Glasgow et al. (2021) predict that FEDAVG's convergence is greatly affected by the data heterogeneity. In particular, when the local gradients on clients become more different from each other (i.e., more data heterogeneity), FEDAVG may require much more communication rounds to converge. These theoretical predictions match with the observations on some datasets with artificially partitioned or synthetic non-IID data (Hsu et al., 2019; Li et al., 2020a; Zhao et al., 2018). However, on many real-world FL training tasks, FEDAVG actually performs extremely well (McMahan et al., 2017; Charles et al., 2021), which appears to be unreasonable based on the existing theory. In fact, many advanced methods aimed at mitigating the negative effects of data heterogeneity performs similar to FEDAVG in real-world FL training tasks. For example, SCAFFOLD (Karimireddy et al., 2020b) needs much fewer communication rounds to converge than FEDAVG in theory and when run on a synthetic dataset. However, SCAFFOLD and FEDAVG are reported to have roughly identical empirical performance on many realistic federated datasets, see the results in (Reddi et al., 2021; Wu et al., 2023; Li et al., 2021).

The above gap between threotical results and practical observations motivates us to think about whether the current theoretical analyses are too pessimistic about FEDAVG, since most of them only focus on the worst cases. It remains as an open problem whether the data heterogeneity modeled by theory and simulated via artificially constructed non-IID datasets matches the heterogeneity in real-world applications. Is it possible that the data heterogeneity in practice has some special properties that allow FEDAVG to enjoy good convergence? The answers to the above questions will serve as important guidelines for the design and evaluation of future federated algorithms.

**Main Contributions.** In this paper, we take the first step towards explaining the practical effectiveness of FEDAVG under a special but realistic hypothesis. In particular, our contributions are as follows.

- By performing experiments on naturally heterogeneous federated datasets, we show that previous theoretical predictions do not align well with practice. FEDAVG can have nearly identical performance on both IID and non-IID versions of these datasets. Thus, previous worst-case analyses may be too pessimistic for such datasets. This is likely because the heterogeneity metric used by existing analyses may be too loose for naturally heterogeneous datasets.

- In order to explain the practical effectiveness of FEDAVG, we propose a *client consensus hypothesis*: the average of local model updates starting from the optimum is close to zero. For smooth and strongly-convex functions, we prove that under this hypothesis, there is no negative impact on the convergence of FEDAVG even with unbounded gradient dissimilarity.

- We further validate that the client consensus hypothesis can hold in many scenarios. We firstly consider a linear regression problem where all the clients have the same conditional probability of label given data, and show that the client consensus hypothesis holds. Besides, we empirically show that natural federated datatsets such as FEMNIST and StackOverflow satisfy this hypothesis. Indeed, data heterogeneity has very limited impact on these datasets.

We would like to clarify that we are *not* trying to provide any practical approach/criterion to predict which datasets FedAvg will converge on. We are just diving deep into the notion of heterogeneity and trying to provide some insights on which notion seems to be more aligned with the behavior of FedAvg in practice.

## 2 Preliminaries and Related Work

**Problem Formulation.** We consider a FL setting with total $M$ clients, where each client $c$ has a local objective function $F_c(\boldsymbol{w})$ defined on its local dataset $\mathcal{D}_c$. The goal of FL training is to minimize a global objective function, defined as a weighted average over all clients:

$$F(\boldsymbol{w}) = \sum_{c=1}^{M} p_c F_c(\boldsymbol{w}) = \mathbb{E}_c[F_c(\boldsymbol{w})], \tag{1}$$

where $p_c$ is the relative weight for client $c$. For the ease of writing, in the rest of this paper, we will use $\mathbb{E}_c[\boldsymbol{a}_c] = \sum_{c=1}^M p_c \boldsymbol{a}_c$ to represent the weighted average over clients.

**Update Rule of FedAvg.** FEDAVG (McMahan et al., 2017) is a popular algorithm to minimize (1) without the need of uploading raw training data. In round $t$ of FEDAVG, client $c$ performs $H$ steps of SGD from the current global model $\boldsymbol{w}^{(t)}$ to a local model $\boldsymbol{w}_c^{(t,H)}$ with a local learning rate $\eta$. Then, at the server, the local model changes are aggregated to update the global model as follows:

$$\boldsymbol{w}^{(t+1)} = \boldsymbol{w}^{(t)} - \alpha \mathbb{E}_c[\boldsymbol{w}^{(t)} - \boldsymbol{w}_c^{(t,H)}]. \tag{2}$$

Here $\alpha$ denotes the server learning rate. Our results in this paper are for the full-device participation case, i.e., when all the clients participate in each round. We discuss how our results can be extended to the partial-device participation case at the end of Section 4.2.

**Theoretical Analysis of FedAvg.** When clients have *homogeneous* data, many works have provided error upper bounds to guarantee the convergence of FEDAVG (also called Local SGD) (Stich, 2019; Yu et al., 2019b; Wang & Joshi, 2018; Zhou & Cong, 2018; Khaled et al., 2020; Li et al., 2019). In these papers, FEDAVG was treated as a method to reduce the communication cost in distributed training. It has been shown that in the stochastic setting, using a proper $H > 1$ in FEDAVG will not negatively influence the dominant convergence rate. Hence FEDAVG can save communication rounds compared to the algorithm with $H = 1$. Later in Woodworth et al. (2020b), the authors compared FEDAVG with the mini-batch SGD baseline, and showed that in certain regimes, FEDAVG provably improves over mini-batch SGD. These upper bounds on FEDAVG was later proved by Glasgow et al. (2021) to be tight and not improvable for general convex functions.

When clients have *heterogeneous* data, in order to analyze the convergence of FEDAVG, it is common to make the following assumption to bound the difference among local gradients.

**Assumption 1** (Bounded Gradient Dissimilarity). *There exists a positive constant $\zeta$ such that $\forall \ \boldsymbol{w} \in \mathbb{R}^d$, the difference between local and global gradients are uniformly bounded:*

$$\mathbb{E}_c \left\| \nabla F_c(\boldsymbol{w}) - \nabla F(\boldsymbol{w}) \right\|^2 \leq \zeta^2. \tag{3}$$

This assumption first appeared in decentralized optimization literature (Lian et al., 2017; Yuan et al., 2016; Assran et al., 2018; Koloskova et al., 2020; Wang et al., 2022), and has been subsequently used in the analysis of FEDAVG and related algorithms (Yu et al., 2019a; Khaled et al., 2020; Karimireddy et al., 2020b; Reddi et al., 2020; Wang et al., 2020a;b; Haddadpour & Mahdavi, 2019; Karimireddy et al., 2020a; Das et al., 2022; Zindari et al., 2023; Crawshaw et al., 2024). Under the bounded gradient dissimilarity assumption, FEDAVG cannot outperform the simple mini-batch SGD baseline unless $\zeta$ is extremely small ($\zeta < 1/T$ where $T$ is the total communication rounds) (Woodworth et al., 2020a); the deterministic version of FEDAVG (i.e., Local GD) has even slower convergence rate than vanilla GD (Khaled et al., 2020). Again, these upper bounds match a lower bound constructed in Glasgow et al. (2021) for general convex functions, suggesting that they are tight in the worst case. In this paper, we do not aim to improve these bounds, which are already tight. Instead, we argue that since the existing analyses only consider the worst case, they may be too pessimistic for practical applications. The data heterogeneity induced by the bounded gradient dissimilarity assumption may be different from the real-world heterogeneity.

Finally, we note that there is another line of works, namely, Malinovskiy et al. (2020); Charles & Konečnỳ (2021); Charles & Rush (2022), using a different analysis technique from the above literature. They showed that FEDAVG (with full client participation) is equivalent to performing gradient descent on a surrogate loss function. However, so far this technique still has some limitations. It can only be applied to deterministic settings with quadratic (or a very special class of) loss functions. Additionally, Wang et al. (2023a) propose a heterogeneity-driven Lipschitz assumption to better capture the effect of local steps, and derive a convergence result for FEDAVG with this assumption. Gu et al. (2023) discuss the reasons why FEDAVG can have better generalization than mini-batch SGD.

**Comparisons of Data Heterogeneity Assumptions.** In Table 1, we summarize some commonly used data heterogeneity assumptions in literature. It is worth highlighting that previous literature, no matter

which heterogeneity measures they used, suggested that only when all local functions are the same or share the same optimum, there are no additional error terms (i.e., $\zeta = 0$) caused by data heterogeneity. However, in our paper, we show that even if local functions are heterogeneous and have different local optima, data heterogeneity can have no negative impact in some regimes (which likely happen in practice). This new result helps us gain a deeper understanding of the great performance of FedAvg observed in practice and cannot be obtained from previous works.

Among all these heterogeneity measures, gradient dissimilarity (at optimum) is considered as the most general one and widely used in literature (Wang et al., 2021). But as we will discuss in the paper, it can be pessimistic in practice. Besides, the gradient diversity assumption used in (Li et al., 2020a; Haddadpour & Mahdavi, 2019) implicitly forces all local functions share the same optimum. So it is much more restrictive than the gradient dissimilarity.

Table 1: Summary of data heterogeneity measures/assumptions used in literature. The above table is adapted from Table 6 in the survey (Kairouz et al., 2019). In the above works, their error upper bounds have the same dependency on $\zeta$, though $\zeta$'s definition is different.

| Name | Definition | Example usage |
|---|---|---|
| grad. dissimilarity | $\mathbb{E}_c\|\nabla F_c(\boldsymbol{w}) - \nabla F(\boldsymbol{w})\| \leq \zeta^2$ | Woodworth et al. (2020a) |
| grad. dissimilarity at opt. | $\mathbb{E}_c\|\nabla F_c(\boldsymbol{w}^*)\|^2 \leq \zeta^2$ | Khaled et al. (2020); Koloskova et al. (2020) |
| grad. diversity | $\mathbb{E}_c\|\nabla F_c(\boldsymbol{w})\|^2 \leq \lambda^2\|\nabla F(\boldsymbol{w})\|^2$ | Li et al. (2020a); Haddadpour & Mahdavi (2019) |
| general grad. diversity | $\mathbb{E}_c\|\nabla F_c(\boldsymbol{w})\|^2 \leq \lambda^2\|\nabla F(\boldsymbol{w})\|^2 + \zeta^2$ | Wang et al. (2020a); Karimireddy et al. (2020b) |
| grad. norm | $\|\nabla F_c(\boldsymbol{w})\|^2 \leq \zeta^2$ | Yu et al. (2019b); Li et al. (2020b) |
| opt. diff | $\|\mathbb{E}_c\boldsymbol{w}_c^* - \boldsymbol{w}^*\|^2 \leq \zeta^2$ | Wang et al. (2023b) |

## 3 Mismatch Between Theory and Practice on FedAvg

In this section, we will introduce the existing convergence analysis in detail and compare the theory with experimental results. We shall show that there is a gap between the theory and practice of FedAvg.

### 3.1 Existing Theoretical Analysis of FedAvg

Besides the bounded gradient dissimilarity assumption (3), the analysis of FedAvg relies on the following common assumptions.

**Assumption 2** (**Lipschitz Smoothness**). *There exists a constant $L$ such that, $\forall \boldsymbol{w}, \boldsymbol{u} \in \mathbb{R}^d, \forall c,$*

$$\|\nabla F_c(\boldsymbol{w}) - \nabla F_c(\boldsymbol{u})\|^2 \leq L\|\boldsymbol{w} - \boldsymbol{u}\|^2 \tag{4}$$

**Assumption 3** (**Local Unbiased Gradient**). *Local stochastic gradient on each client $c$ is unbiased with expectation $\nabla F_c(\boldsymbol{w})$ and bounded variance $\sigma^2$.*

Using the above assumptions, previous works derived an upper bound for the optimization error with non-convex objective functions. We take the following theorem from (Jhunjhunwala et al., 2022) as an example.

**Theorem 1.** *Under Assumptions 1 to 3, if FedAvg's learning rates satisfy $\eta \leq 1/8LH, \alpha \leq 1/24LH$, then the global gradient norm $\min_t \mathbb{E}\|\nabla F(\boldsymbol{w}^{(t)})\|^2$ can be upper bounded by*

$$\mathcal{O}\left(\frac{F(\boldsymbol{w}^{(0)}) - F^*}{\alpha\eta HT}\right) + \mathcal{O}\left(\frac{\alpha\eta L\sigma^2}{M} + \eta^2 L^2 H\sigma^2\right) + \mathcal{O}\left(\eta^2 L^2 H^2 \zeta^2\right). \tag{5}$$

Theorem 1 shows that when the learning rates $\alpha, \eta$ are *fixed* and the communication round $T$ is limited, data heterogeneity always introduces an addition term $\mathcal{O}(\zeta^2)$ to the optimization error bound. It is worth noting that, with an optimized learning rate, the above upper bound matches a lower bound constructed in Glasgow et al. (2021) for general convex functions, suggesting that it is tight. In the worst case, the convergence of

FEDAVG will become worse compared to the homogeneous setting. Although some papers argued that the data heterogeneity only influences the higher order terms when using optimized learning rates (Yu et al., 2019a; Khaled et al., 2020), this conclusion only holds asymptotically w.r.t. $T$. Also, in some special cases, FEDAVG's convergence rate with data heterogeneity can be substantially slower. For instance, when there is no stochastic noise $\sigma = 0$, FEDAVG in homogeneous setting can achieve a rate of $T^{-1}$ but with heterogeneous data, the rate degrades to $T^{-2/3}$ (Woodworth et al., 2020a; Wang et al., 2021; Jhunjhunwala et al., 2022).

### 3.2 Empirical Observations on FedAvg

**Results on Artificial Non-IID Datasets.** In order to corroborate the above theoretical results, most of previous works constructed datasets with artificially high data heterogeneity. Given a common benchmark classification dataset (such as MNIST, CIFAR-10/100), researchers simulated the data heterogeneity by assigning different class distributions to clients. For example, each client may only hold one or very few classes of data (Zhao et al., 2018), or has data for all classes but the amount of each class is randomly drawn from a Dirichlet distribution (Hsu et al., 2019). On these datasets, the empirical convergence of FEDAVG is greatly impacted and its final accuracy is much lower than the one in the centralized training setting. As shown in Zhao et al. (2018), FEDAVG's final test accuracy on non-IID versions of CIFAR-10 can be $10 - 40\%$ lower than its on the standard IID version.

**Results on Natural Non-IID Datasets.** While the negative results on artificial non-IID datasets are widely cited to claim FEDAVG suffers due to data heterogeneity, we doubt whether the results are representative and general enough to cover all practical applications. Is it possible that there are some scenarios where the data heterogeneity has benign effects and has different characteristics from the heterogeneity simulated through these originally IID datasets?

We conduct some experiments on StackOverflow, a naturally non-IID split dataset for next-word prediction. Each client in the dataset corresponds to a unique user on the Stack Overflow site. The data of each client consists of questions and answers written by the corresponding user. More details about the experimental setup, such as optimizer and learning rate choices, can be found in the Appendix. From the naturally heterogeneous StackOverflow dataset, we create its IID version by aggregating and shuffling the data from all clients, and then re-assigning the IID data back to clients. Surprisingly, the results in Figure 1 show that the convergence of FEDAVG is nearly identical (with limited communication rounds) on the new IID dataset and its original non-IID version. This observation contradicts the conventional wisdom that data heterogeneity always adds an additional error to the optimization procedure, as well as the empirical results on *artificial non-IID* datasets. There is indeed a gap between the theory and practice of FEDAVG.

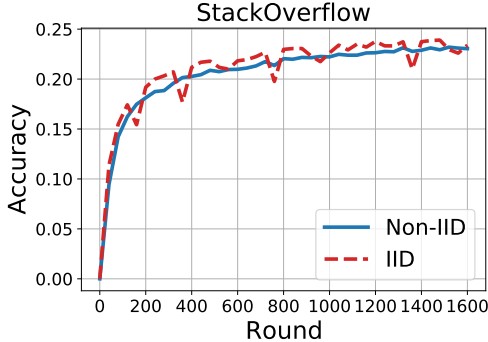

Figure 1: **Results on the naturally non-IID dataset StackOverflow.** The IID version of StackOverflow is created via aggregating and shuffling all the data from clients. We observe that FEDAVG achieves roughly the same convergence in both IID and non-IID settings.

Experimental results from some previous papers serve as additional evidence to support our claim. For instance, FEDAVG converges significantly faster than FEDSGD on Shakespeare and StackOverflow datasets

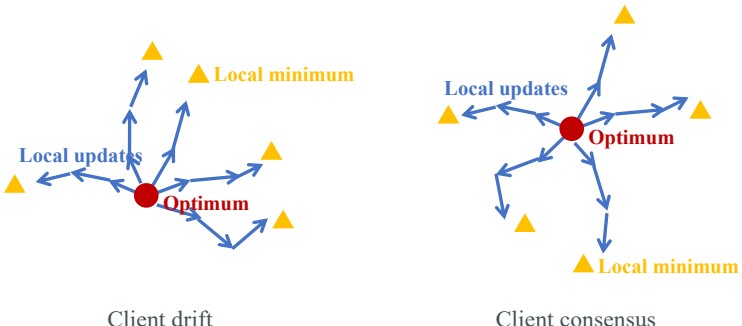

Figure 2: Comparison between client drift and client consensus. (Left) Previous works focus on the worst case where the average of clients' local updates (or drifts) drives the global model away from the optimum; (Right) Our client consensus hypothesis states that the average local drifts at the optimum is very small or close to zero. This structured heterogeneity does not impact the convergence of FEDAVG.

even with strong heterogeneity (McMahan et al., 2017; Charles et al., 2021). Even though a more sophisticated method like SCAFFOLD does better on artificial datasets like EMNIST, it does not outperform FEDAVG on many naturally heterogeneous datasets like FEMNIST, StackOverflow, etc., as shown in (Reddi et al., 2021; Wu et al., 2023; Li et al., 2021). Explanations to these observations still remain mysteries.

## 4   Client Consensus Hypothesis

In previous sections, we saw that there is a gap between the theory and practice of FEDAVG. While theory predicts that FEDAVG performs worse in the presence of data heterogeneity, we did not observe this on naturally non-IID datasets. Also, many advanced methods designed to tackle data heterogeneity problem do not outperform FEDAVG. In order to explain the above phenomenon, in this section, we propose the client consensus hypothesis. Many realistic federated datasets may satisfy this special property such that data heterogeneity has very limited impacts on the convergence of FEDAVG.

### 4.1   Key Insights

FEDAVG is commonly viewed as a perturbed version of vanilla SGD. To see this, we can consider the accumulated local updates on each client as a pseudo-gradient, which is an approximation of the batch client gradient. When clients have heterogeneous data, the average of the pseudo-gradients across all clients can be very different from the original global gradient. Especially, when the global model approaches the optimum (or a stationary point) with a constant learning rate, since the average of pseudo-gradients is not zero, the global model cannot stay and may drift away towards a different point. This is referred to as client drift problem in literature (Karimireddy et al., 2020b). An illustration is provided in Figure 2.

The above insight is true in the worst case. But it is possible that real-world FL datasets are far away from this worst case. In this paper, we make a hypothesis about a special class of data heterogeneity. In particular, clients can still have drastically different data distributions and local minima. But, at the global optimum (or stationary point), clients' local updates (i.e., pseudo-gradients) cancel out with each other and hence, the average pseudo-gradient at the optimum is or close to zero. As a consequence, when the global model reach the global optimum, it can stay there and will not drift away. That is, clients reach some kind of consensus at the optimum. For this reason, we name the hypothesis as *client consensus hypothesis*. We will show it in Section 4.2 that this special class of non-IIDness has no negative impacts on the convergence of FEDAVG. Therefore, the hypothesis can serve as a possible explanation of the effectiveness of FEDAVG.

More formally, we define the deterministic pseudo-gradient for client $c$ as follows:

$$\mathcal{G}_c(\boldsymbol{w}) \triangleq \frac{1}{\eta H}(\boldsymbol{w} - \boldsymbol{w}_{c,\mathrm{GD}}^{(H)}) \tag{6}$$

where $\boldsymbol{w}_{c,\text{GD}}^{(H)}$ denotes the locally trained model at client $c$ after performing $H$ steps of GD from $\boldsymbol{w}$ using a local learning rate $\eta$. Given this definition, the client consensus hypothesis is formally stated below.

**Assumption 4** (**Client Consensus Hypothesis**). *On real-world federated datasets, for the values of $\eta, H$ used in* FEDAVG*, the average pseudo-gradient at the optimum (i.e., average client model drift at the optimum)*

$$\rho \triangleq \left\| \mathbb{E}_c[\mathcal{G}_c(\boldsymbol{w}^*)] \right\|$$

*is very small or close to zero, where $\boldsymbol{w}^*$ is the global optimum or stationary point of the global objective* (1).

Note that the Client Consensus Hypothesis is satisfied whenever FEDAVG converges with a constant learning rate. Thus, in a sense, it is the minimal assumption under which one can expect to prove improved convergence rates for FEDAVG. $\rho$ can be interpreted as the *average drift at optimum*; this quantity is akin to a *heterogeneity metric* and our theoretical results will be in terms of it.

**Connections to Existing Analysis with Bounded Gradient Dissimilarity.** Here, we discuss the connections between client consensus hypothesis and previous convergence analysis with bounded gradient dissimilarity assumption. As we mentioned before, FEDAVG can be viewed as a perturbed version of vanilla SGD. This alternative view is also critical in the convergence analysis. A key step in previous FEDAVG analysis is to bound the perturbations, i.e., the difference between the average pseudo-gradient $\mathbb{E}_c[\mathcal{G}_c]$ and the batch global gradient $\nabla F(\boldsymbol{w}) = \mathbb{E}_c[\nabla F_c(\boldsymbol{w})]$. Previous works always upper bound this difference using Jensen's inequality as follows:

$$\|\mathbb{E}_c[\mathcal{G}_c(\boldsymbol{w}) - \nabla F_c(\boldsymbol{w})]\|^2 \leq \mathbb{E}_c\|\mathcal{G}_c(\boldsymbol{w}) - \nabla F_c(\boldsymbol{w})\|^2. \tag{7}$$

Then, the right-hand-side (RHS) can be further bounded using the gradient dissimilarity assumption. However, we note that the above upper bound omits the correlations among different clients. While the RHS (i.e., average of squared $\ell_2$ differences) can be large or unbounded, the LHS (i.e., squared $\ell_2$ norm of the average difference) can be small or just zero, especially at the optimum $\boldsymbol{w}^*$. Therefore, using (7) in the analysis may result in a pessimistic estimate.

Instead, in this paper, the client consensus hypothesis states that the LHS of (7) at the optimum is close to, if not, zero. Using the hypothesis and our new analysis, we no longer need to provide an upper bound of the RHS of (7) for all points. As a consequence, the new analysis does not require the gradient dissimilarity to be bounded.

## 4.2 Convergence of FedAvg under Client Consensus Hypothesis

In this subsection, we will provide a convergence analysis for FEDAVG under the client consensus hypothesis (Assumption 4) for strongly-convex loss functions. In particular, we show that when $\rho = 0$ in Assumption 4, *data heterogeneity has no negative impact* on the convergence of FEDAVG.

Besides the pseudo-gradient defined in (6), we need to define its stochastic version:

$$\hat{\mathcal{G}}_c(\boldsymbol{w}) \triangleq \frac{1}{\eta H}(\boldsymbol{w} - \boldsymbol{w}_c^{(H)}) \tag{8}$$

where $\boldsymbol{w}_c^{(H)}$ denotes the locally trained model at client $c$ using $H$ steps of SGD with learning rate $\eta$, starting from $\boldsymbol{w}$. Let us define $\mathcal{G} := \mathbb{E}_c[\mathcal{G}_c]$ and $\hat{\mathcal{G}} := \mathbb{E}_c[\hat{\mathcal{G}}_c]$. Then, we have the following result.

**Theorem 2.** *Under Assumptions 2 to 4, when each local objective function is $\mu$-strongly convex and the learning rates satisfy $\alpha \leq 1/4, \eta \leq \min\{1/L, 1/\mu H\}$, after $T$ rounds of* FEDAVG*, we have*

$$\mathbb{E}\left\| \boldsymbol{w}^{(T)} - \boldsymbol{w}^* \right\|^2 \leq (1 - \frac{1}{2}\alpha\eta H\mu)^T \left\| \boldsymbol{w}^{(0)} - \boldsymbol{w}^* \right\|^2 + \frac{2\alpha\eta H}{\mu} \max_{\boldsymbol{w}} \text{Var}[\hat{\mathcal{G}}(\boldsymbol{w})]$$

$$+ \frac{20}{\mu^2} \max_{\boldsymbol{w}} \mathbb{E}_c \left\| \delta_c(\boldsymbol{w}) \right\|^2 + \frac{20\rho^2}{\mu^2} \tag{9}$$

*where $\mathbb{E}[\cdot], \text{Var}[\cdot]$ are taken with respect to random noise in stochastic local updates, and $\delta_c(\boldsymbol{w}) = (\boldsymbol{w}_{c,GD}^{(H)} - \mathbb{E}[\boldsymbol{w}_c^{(H)}])/(\eta H)$ denotes the iterate bias between local GD and local SGD on client $c$.*

From Theorem 2, we observe that the stochastic noise during local updates influences the second and the third terms on the right-hand-side of (9). The upper bounds of these two terms only depend on the dynamics of SGD, which has been well understood in literature. Specifically, in Khaled et al. (2020), the authors show that $\mathbb{E}\|\boldsymbol{w}^{(H)} - \mathbb{E}[\boldsymbol{w}^{(H)}]\|^2 \leq 2\eta^2 H \sigma^2$. As a consequence, we directly obtain that

$$\text{Var}[\hat{\mathcal{G}}(\boldsymbol{w})] = \frac{\mathbb{E}\left\|\boldsymbol{w}_c^{(H)} - \mathbb{E}[\boldsymbol{w}_c^{(H)}]\right\|^2}{\eta^2 H^2 M} \leq \frac{2\sigma^2}{MH}. \tag{10}$$

As for the iterate bias, one can obtain

$$\mathbb{E}_c \left\|\delta_c(\boldsymbol{w})\right\|^2 \leq \eta^2 L^2 \sigma^2 (H-1), \tag{11}$$

the proof of which is provided in the Appendix. After substituting (10) and (11) into (9) and optimizing the learning rates, we can obtain the following convergence rate for FEDAVG.

**Corollary 1** (**Convergence Rate for Strongly Convex Functions**). *In the same setting as Theorem 2, when $\alpha = 1/4, \eta = \mathcal{O}(1/\mu HT)$, the convergence rate of* FEDAVG *is[1]*

$$\mathbb{E}\left\|\boldsymbol{w}^{(T)} - \boldsymbol{w}^*\right\|^2 = \widetilde{\mathcal{O}}\left(\frac{\sigma^2}{MHT} + \frac{\sigma^2}{HT^2} + \rho^2\right). \tag{12}$$

*If clients perform local GD instead of local SGD, then when $\eta = \min\{1/L, 1/(\mu H)\}$, we have*

$$\left\|\boldsymbol{w}^{(T)} - \boldsymbol{w}^*\right\|^2 = \mathcal{O}\left(\exp\left(-\frac{T}{16\kappa}\min\{\kappa, H\}\right) + \rho^2\right) \tag{13}$$

*where $\kappa = L/\mu$ denotes the condition number.*

**Effects of Data Heterogeneity.** In the special regime of $\rho \approx 0$, i.e., when the client consensus hypothesis holds, Theorem 2 and Corollary 1 state that data heterogeneity does not have any negative impact on the convergence of FEDAVG. However, in previous works based on gradient dissimilarity, even if $\rho = 0$, there is an additional error caused by the positive gradient dissimilarity. Compared to the centralized setting where $M = 1$, (12) suggests that FEDAVG can provide linear speedup due to the usage of multiple workers.

**Extensions.** The above analysis can be extended to various settings. Below we provide several examples.

(1) *Client Sampling*: If we consider client sampling in FEDAVG, then only the variance term in (9) will change while the other terms will be unaffected. One can obtain new convergence guarantees by analyzing the variance of different sampling schemes and then simply substituting them into (9). Standard techniques to analyze client sampling (Yang et al., 2020) can be directly applied.

(2) *Third-order Smoothness*: When the local objective functions satisfy third-order smoothness ($\left\|\nabla^2 F_c(\boldsymbol{w}) - \nabla^2 F_c(\boldsymbol{u})\right\| \leq Q \left\|\boldsymbol{w} - \boldsymbol{u}\right\|$), the bound of the iterate bias $\delta(\boldsymbol{w})$ can be further improved while all other terms remain unchanged. According to Glasgow et al. (2021), one can obtain $\|\delta(\boldsymbol{w})\| \leq \frac{1}{4}\eta^2 HQ\sigma^2$. In the special case when local objective functions are quadratic, we have $Q = 0$. That is, there is no iterate bias. As a consequence, the convergence rate of FEDAVG can be significantly improved.

Moreover, a result for the general convex case is provided in the Appendix. Analysis for non-convex functions requires a different framework and is non-trivial. So we leave it for future work.

**Comparison with Previous Works.** In Table 2, we summarize the convergence rates of FEDAVG in different papers. All previous results depend on the gradient dissimilarity upper bound $\zeta$, which is generally large for heterogeneous data settings. However, in our result, under client consensus hypothesis, we show that the effects of data heterogeneity can be measured by average drift at optimum $\rho$, which can be close to zero even in presence of strong data heterogeneity, as we showed in the quadratic example and experiments on FEMNIST and StackOverflow. When the average drift at optimum is zero, we can get an improved convergence rate compared to existing results.

---

[1]$\widetilde{\mathcal{O}}(.)$ hides log factors.

Table 2: Comparison with existing results for strongly convex objectives functions with deterministic local updates. In the table, the error is measured by the distance to the optimum $\|\boldsymbol{w} - \boldsymbol{w}^*\|^2$, and $\kappa = L/\mu$ is the condition number. Also, we omit logarithmic factors. Compared to previous results, we show that in the considered setting: (i) FEDAVG enjoys linear convergence to the global optimum, and (ii) the multiple local steps of FEDAVG mitigate the impact of of ill-conditioning (high condition number).

| Algorithm | Worst-case error | Comm. rounds to attain $\epsilon$ error when $\rho = 0$ |
|---|---|---|
| GD | $\exp(-T/\kappa)$ | $\mathcal{O}(\kappa \log(1/\epsilon))$ |
| FEDAVG (Koloskova et al., 2020) | $\zeta^2/T^2$ | $\mathcal{O}(1/\epsilon^2)$ |
| FEDAVG (Woodworth et al., 2020a) | $1/(HT + H^2T^2) + \zeta^2/T^2$ | $\mathcal{O}(1/\epsilon^2)$ |
| FEDAVG (Ours) | $\exp(-T \min\{1, H/\kappa\}) + \rho^2$ | $\mathcal{O}(\max\{1, \kappa/H\} \log(1/\epsilon))$ |

## 5 Validating the Client Consensus Hypothesis

Here we will present some evidence to show that the client consensus hypothesis is realistic and practical; we do so (a) theoretically, in a linear regression setting, and (b) empirically, on some naturally split datasets such as FEMNIST and StackOverflow.

### 5.1 Linear Regression Example

**Intuition on When Client Consensus Hypothesis Holds.** One of the key insights of the client consensus hypothesis is that clients do reach some consensus and a single global model can work reasonably well for all clients though they have heterogeneous data. Inspired by this, we assume that all clients have the same conditional probability of the label given data, i.e., $p_c(y|\boldsymbol{x}) = p(y|\boldsymbol{x}), \forall c$, where $\boldsymbol{x}, y$ denote the input data and its label, respectively. In this case, clients still have heterogeneous data distributions, as they may have drastically different $p_c(\boldsymbol{x})$ and $p_c(\boldsymbol{x}, y)$. However, the client's updates should not conflict with each other, as the learning algorithms tend to learn the same $p(y|\boldsymbol{x})$ on all clients.

Let us study a concrete *linear regression* setting satisfying the above property. Specifically, we assume that the label of the $i^{\text{th}}$ data sample on client $c$ is generated as follows:

$$y_{c,i} = \langle \boldsymbol{w}^*, \boldsymbol{x}_{c,i} \rangle + \epsilon_{c,i}, \tag{14}$$

where $\boldsymbol{w}^* \in \mathbb{R}^d$ denotes the optimal model, and $\epsilon_{c,i} \sim \mathcal{P}_\epsilon$ is a zero-mean random noise and independent from $\boldsymbol{x}_{c,i}$ (this is a common assumption in statistical learning). We also assume that all $\|\boldsymbol{x}_{c,i}\|$ have bounded variance. Note that both $\boldsymbol{w}^*$ and $\mathcal{P}_\epsilon$ are the same across all the clients, i.e., all clients have the same label generation process and hence, the same conditional probability $p(y|\boldsymbol{x})$. Our goal is to find the optimal model $\boldsymbol{w}^*$ given a large amount of clients with *finite* data samples (which is a common cross-device FL setting (Kairouz et al., 2019)). We use the squared loss function which makes our objective function *quadratic*; specifically, it is:

$$
\begin{aligned}
F_c(\boldsymbol{w}) &= \frac{1}{n} \sum_{i=1}^{n} \frac{1}{2} \left( y_{c,i} - \boldsymbol{w}^\top \boldsymbol{x}_{c,i} \right)^2 \\
&= \frac{1}{2}(\boldsymbol{w} - \boldsymbol{w}^*)\boldsymbol{A}_c(\boldsymbol{w} - \boldsymbol{w}^*) - \boldsymbol{b}_c^\top (\boldsymbol{w} - \boldsymbol{w}^*) + \text{const.},
\end{aligned} \tag{15}
$$

where $\boldsymbol{A}_c = \sum_{i=1}^{n} \boldsymbol{x}_{c,i} \boldsymbol{x}_{c,i}^\top / n$, $\boldsymbol{b}_c = \sum_{i=1}^{n} \epsilon_{c,i} \boldsymbol{x}_{c,i} / n$. The minimizer of local objective $F_c(\boldsymbol{w})$ is $\boldsymbol{w}_c^* = \boldsymbol{w}^* + \boldsymbol{A}_c^{-1}\boldsymbol{b}_c$, which is different from the global minimizer $\boldsymbol{w}^*$ as $\boldsymbol{b}_c \neq 0$.

**Client Consensus Hypothesis.** In this problem, we can show that client consensus hypothesis holds. In particular, one can show that the average pseudo-gradient at the optimum $\rho$ is

$$\mathbb{E}_c[\mathcal{G}_c(\boldsymbol{w}^*)] = \mathbb{E}_c \left[ \frac{1}{H} \sum_{h=0}^{H-1} \left[ \boldsymbol{I} - (\boldsymbol{I} - \eta \nabla^2 \boldsymbol{A}_c)^h \right] \boldsymbol{b}_c \right]. \tag{16}$$

Due to the independence of $\epsilon_{c,i}$ and $\boldsymbol{x}_{c,i}$, for any choice of $H$, $\rho = \|\mathbb{E}_c[\mathcal{G}_c(\boldsymbol{w}^*)]\| \to 0$ almost surely when the number of clients $M \to \infty$.

**Unbounded Gradient Dissimilarity.** In contrast, if we check the gradient dissimilarity, we have:

$$\mathbb{E}_c \|\nabla F_c(\boldsymbol{w}^*) - \nabla F(\boldsymbol{w}^*)\|^2 = \mathbb{E}_c \|\boldsymbol{b}_c\|^2. \tag{17}$$

Observe that $\epsilon$ can have extremely large variance such that the gradient dissimilarity bound $\zeta$ is arbitrarily large. As a result, existing analyses, which rely on the bounded gradient dissimilarity, may predict that FEDAVG is much worse than its non-local counterparts. However, by simple manipulations on the update rule of FEDAVG, one can prove the following theorem.

**Theorem 3.** *Suppose that the weighting of the clients is uniform, and each client has a small finite amount of data. For the above linear regression setting, the iterates of Local GD (i.e., deterministic version of* FEDAVG*) satisfy the following equation almost surely as the number of clients goes to infinity:*

$$\boldsymbol{w}^{(T)} - \boldsymbol{w}^* = \left[\mathbb{E}_c \left[(\boldsymbol{I} - \eta \boldsymbol{A}_c)^H\right]\right]^T (\boldsymbol{w}^{(0)} - \boldsymbol{w}^*). \tag{18}$$

The proof is relegated to the Appendix. From Theorem 3, it is clear that if the learning rate $\eta$ is properly set such that $(\boldsymbol{I} - \eta \boldsymbol{A}_c)$ is positive definite, then performing more local updates (larger $H$) will lead to faster *linear* convergence rate $\mathcal{O}(\exp(-T))$ to the global optimum $\boldsymbol{w}^*$. That is, local GD is strictly better than vanilla GD. However, previous papers based on gradient dissimilarity will get a substantially slower rate of $\mathcal{O}(1/T^2)$. In this example, while gradient dissimilarity can be arbitrarily large, data heterogeneity does not have any negative impacts on FEDAVG as the client consensus hypothesis holds.

## 5.2 Experiments on Naturally Non-IID Datasets

In this subsection, we empirically show that the proposed client consensus hypothesis approximately holds across multiple practical training tasks.

In Figure 3, we first run mini-batch SGD on Federated EMNIST (FEMNIST) (McMahan et al., 2017) and StackOverflow Next Word Prediction datasets (Reddi et al., 2019) to obtain an approximation for the optimal model $\boldsymbol{w}^*$. Then we evaluate the average drift at optimum $\rho = \|\mathbb{E}_c B_c(\boldsymbol{w}^*)\|$ and its upper bound as given in (7) on these datasets. We summarize the important observations below.

- As shown in Figures 3a and 3b, the average drift (or pseudo-gradient) at the optimum, i.e. $\rho$, is very close to zero on naturally non-IID datasets FEMNIST and StackOverflow. The proposed client consensus hypothesis is true on these two realistic datasets.

- From Figures 3a and 3b, we also observe that there is a large gap beteween $\rho$ and its upper bound as given in (7). This suggests that previous analyses using this upper bound can be loose.

- We run the same set of experiments on a non-IID CIFAR-100 dataset. Figure 3c shows that on this artificial dataset, the client consensus hypothesis no longer holds. The average drfit at optimum $\rho$ is pretty close to its upper bound and far from zero. This is the scenario where FEDAVG fails with heterogeneous data and faces the client drift problem.

The drastically different observations on FEMNIST, StackOverflow and artificial CIFAR-100 highlight that there are various kinds of data heterogeneity. For the heterogeneity satisfying client consensus hypothesis, it may have very limited impacts on the convergence of FEDAVG.

Furthermore, we run FEDAVG on FEMNIST dataset following the same setup as (Reddi et al., 2021) and check the difference between the average of pseudo-gradient and the true batch gradient at several intermediate points on the optimization trajectory. For each point, we let clients perform local GD with the same local learning rate for multiple steps. As shown in Figure 4, we observe a significant gap between the norm of average differences (red lines) and its upper bound (7) (i.e., the average of $\ell_2$ difference, blue lines). Especially, at the 50$^{\text{th}}$ and 100$^{\text{th}}$ rounds, the upper bound is about 10 times larger. These observations suggest that the upper bounds based on the gradient dissimilarity are loose in practice.

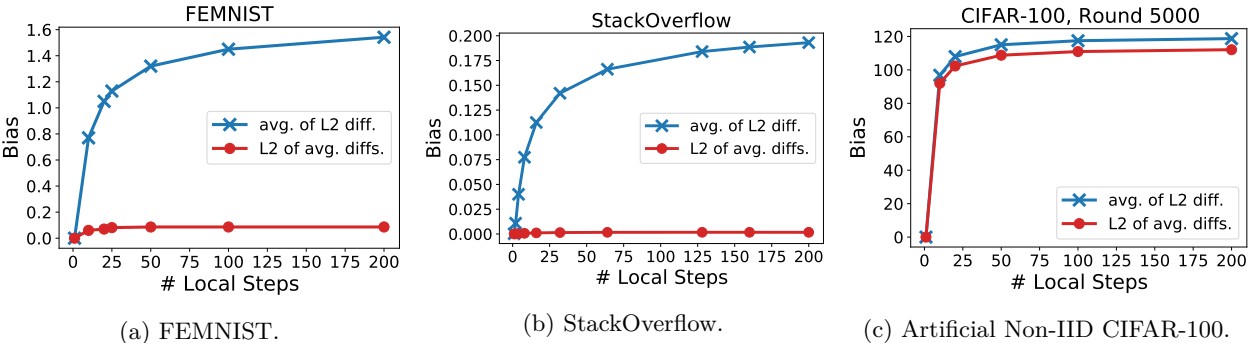

(a) FEMNIST.      (b) StackOverflow.    (c) Artificial Non-IID CIFAR-100.

Figure 3: Difference between the average pseudo-gradient and the global gradient at the optimal point $\boldsymbol{w}^*$ on three different datasets. We observe that the norm of the average gradient differences at $\boldsymbol{w}^*$ (red line) nearly remain zero on all natrual non-IID datasets but its upper bound used in previous analyses (7) (blue lines) slowly become larger when $H$ increases. On a artificial non-IID CIFAR-100 dataset, these two values are pretty close to each other.

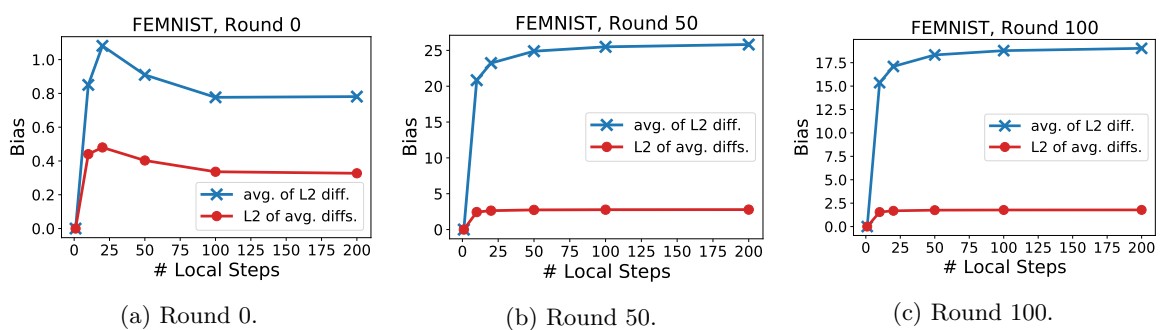

(a) Round 0.       (b) Round 50.      (c) Round 100.

Figure 4: Difference between the average pseudo-gradient and the global gradient on a FedAvg's optimization trajectory. There is a significant gap between the average gradient differences (red line) and its upper bound (blue lines). Both of them increase and then saturate when increasing $H$.

## 6 Conclusions

In this paper, we aim to bridge the gap between theory and practice of the popular FedAvg algorithm. We found that previous analyses based on the bounded gradient dissimilarity assumption can be too pessimistic for practical applications. On some natural federated datasets, FedAvg may have identical performance on both IID and non-IID settings. In order to explain this phenomenon, we introduced the client consensus hypothesis and formally proved that under this hypothesis, data heterogeneity does not exacerbate the convergence of FedAvg. More importantly, we show that this hypothesis holds for a linear regression problem and many practical federated datasets, including FEMNIST and StackOverflow. So the proposed hypothesis is realistic and can better explain the empirical success of FedAvg.

Given the above contributions, several future directions are ripe for exploration. Our client consensus hypothesis is expressed in terms of a quantity ($\rho$) that is akin to a heterogeneity metric; it would be interesting to come up with a formal metric that applies to all settings and can replace the existing loose gradient dissimilarity metric. Perhaps such metrics could also be helpful in guiding the design principles for FL algorithms. Besides, it may be also worthwhile to design a more practical criterion to check whether a dataset satisfies the client consensus hypothesis.

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

## A  Experimental Details

On FEMNIST, StackOverflow, and CIFAR-100 datasets, we strictly follow the training setup given in Reddi et al. (2020). Both models are neural networks and hence the objective functions are non-convex. In Figure 3, we used a trained model (obtained after training with mini-batch SGD) as an approximation of the global optimum. Then, a large set of clients are selected to perform local model training from the optimum to calculate the gradient bias. Details on the local training can be found in the following table.

Table 3: Details on local training in Figure 3.

| Dataset | Model | Loss function | # of clients | Local optimizer | Local learning rate |
|---|---|---|---|---|---|
| FEMNIST | ConvNet | Cross-Entropy | 500 | GD | 0.1 |
| StackOverflow | LSTM | Cross-Entropy | 1000 | GD | 0.5 |
| CIFAR-100 | ConvNet | Cross-Entropy | 200 | GD | 0.5 |

## B  Proof of Theorem 2

### B.1  Preliminaries

In this subsection, we will first introduce several useful lemmas, which relate to the properties of the deterministic pseudo-gradients. Before diving into the proof details, we would like to first introduce a lemma, which will be frequently used in the subsequent sections.

**Lemma 1** (**Mean Value Theorem**). *Suppose function $F$ is twice differentiable, then*

$$\nabla F_c(\boldsymbol{w}) - \nabla F_c(\boldsymbol{u}) = \boldsymbol{A}_c(\boldsymbol{w}, \boldsymbol{u}) \cdot (\boldsymbol{w} - \boldsymbol{u}) \tag{19}$$

*where $\boldsymbol{A}_c(\boldsymbol{w}, \boldsymbol{u}) = \int_0^1 \nabla^2 F_c(\boldsymbol{u} + s(\boldsymbol{w} - \boldsymbol{u}))ds$. If $\mu \preceq \nabla^2 F_c \preceq L$, then it follows that $\mu \preceq \boldsymbol{A}_c(\boldsymbol{w}, \boldsymbol{u}) \preceq L$ for any $\boldsymbol{w}, \boldsymbol{u}$.*

**Lemma 2** (**Convexity, Smoothness & Co-coercivity**). *When each local objective function $F_c$ is $L$-Lipschitz smooth and $\mu$-strongly convex, for any $\boldsymbol{w}, \boldsymbol{u} \in \mathbb{R}^d$, we have*

$$\widetilde{\mu} \|\boldsymbol{w} - \boldsymbol{u}\|^2 \leq \langle \mathcal{G}(\boldsymbol{w}) - \mathcal{G}(\boldsymbol{u}), \boldsymbol{w} - \boldsymbol{u} \rangle \leq \widetilde{L} \|\boldsymbol{w} - \boldsymbol{u}\|^2, \tag{20}$$

$$\|\mathcal{G}(\boldsymbol{w}) - \mathcal{G}(\boldsymbol{u})\|^2 \leq \widetilde{L} \langle \mathcal{G}(\boldsymbol{w}) - \mathcal{G}(\boldsymbol{u}), \boldsymbol{w} - \boldsymbol{u} \rangle, \tag{21}$$

*where $\widetilde{\mu} = [1 - (1 - \eta\mu)^H]/(\eta H)$ and $\widetilde{L} = [1 + (1 - \eta\mu)^H]/(\eta H)$.*

*Proof.* Let us first focus on the pseudo-gradient on a specific client $c$. According to the definition of pseudo-gradient, we have

$$\eta H[\mathcal{G}_c(\boldsymbol{w}) - \mathcal{G}_c(\boldsymbol{u})] = \boldsymbol{w} - \boldsymbol{u} - (\boldsymbol{w}_c^{(H)} - \boldsymbol{u}_c^{(H)}) \tag{22}$$

$$= \boldsymbol{w} - \boldsymbol{u}$$
$$\quad - [\boldsymbol{w}_c^{(H-1)} - \boldsymbol{u}_c^{(H-1)} - \eta(\nabla F_c(\boldsymbol{w}_c^{(H-1)}) - \nabla F_c(\boldsymbol{u}_c^{(H-1)}))] \tag{23}$$

$$= \boldsymbol{w} - \boldsymbol{u} - (I - \eta\boldsymbol{D}_c^{(H-1)})(\boldsymbol{w}_c^{(H-1)} - \boldsymbol{u}_c^{(H-1)}) \tag{24}$$

where (24) follows from Lemma 1 and $\boldsymbol{D}_c$ is a symmetric matrix satisfying $\mu \preceq \boldsymbol{D}_c \preceq L$. Repeating the above procedure, we can obtain that

$$\eta H[\mathcal{G}_c(\boldsymbol{w}) - \mathcal{G}_c(\boldsymbol{u})] = \boldsymbol{w} - \boldsymbol{u} - \prod_{k=0}^{H-1}(\boldsymbol{I} - \eta\boldsymbol{D}_c^{(k)})(\boldsymbol{w} - \boldsymbol{u}) \tag{25}$$

$$= \left[\boldsymbol{I} - \prod_{k=0}^{H-1}(\boldsymbol{I} - \eta\boldsymbol{D}_c^{(k)})\right](\boldsymbol{w} - \boldsymbol{u}). \tag{26}$$

As a consequence, we have

$$\eta H \langle \mathcal{G}_c(\boldsymbol{w}) - \mathcal{G}_c(\boldsymbol{u}), \boldsymbol{w} - \boldsymbol{u} \rangle = \|\boldsymbol{w} - \boldsymbol{u}\|^2 - \left\langle \prod_{k=0}^{H-1} (\boldsymbol{I} - \eta \boldsymbol{D}_c^{(k)})(\boldsymbol{w} - \boldsymbol{u}), \boldsymbol{w} - \boldsymbol{u} \right\rangle. \tag{27}$$

Note that, due to Cauchy–Schwarz inequality,

$$\left| \left\langle \prod_{k=0}^{H-1} (\boldsymbol{I} - \eta \boldsymbol{D}_c^{(k)})(\boldsymbol{w} - \boldsymbol{u}), \boldsymbol{w} - \boldsymbol{u} \right\rangle \right| \leq \prod_{k=0}^{H-1} \left\| \boldsymbol{I} - \eta \boldsymbol{D}_c^{(k)} \right\| \|\boldsymbol{w} - \boldsymbol{u}\|^2 \tag{28}$$

$$\leq (1 - \eta \mu)^H \|\boldsymbol{w} - \boldsymbol{u}\|^2. \tag{29}$$

That is,

$$-(1 - \eta \mu)^H \|\boldsymbol{w} - \boldsymbol{u}\|^2 \leq \left\langle \prod_{k=0}^{H-1} (\boldsymbol{I} - \eta \boldsymbol{D}_c^{(k)})(\boldsymbol{w} - \boldsymbol{u}), \boldsymbol{w} - \boldsymbol{u} \right\rangle \leq (1 - \eta \mu)^H \|\boldsymbol{w} - \boldsymbol{u}\|^2. \tag{30}$$

It follows that,

$$\frac{1 - (1 - \eta \mu)^H}{\eta H} \|\boldsymbol{w} - \boldsymbol{u}\|^2 \leq \langle \mathcal{G}_c(\boldsymbol{w}) - \mathcal{G}_c(\boldsymbol{u}), \boldsymbol{w} - \boldsymbol{u} \rangle \leq \frac{1 + (1 - \eta \mu)^H}{\eta H} \|\boldsymbol{w} - \boldsymbol{u}\|^2. \tag{31}$$

Taking the average over all clients, we complete the proof of (20).

Next, we are going to prove (21). Note that

$$\left\| \boldsymbol{w}_c^{(H)} - \boldsymbol{u}_c^{(H)} \right\| = \left\| \prod_{k=0}^{H-1} (\boldsymbol{I} - \eta \boldsymbol{D}_c^{(k)})(\boldsymbol{w} - \boldsymbol{u}) \right\| \tag{32}$$

$$\leq \prod_{k=0}^{H-1} \left\| \boldsymbol{I} - \eta \boldsymbol{D}_c^{(k)} \right\| \|\boldsymbol{w} - \boldsymbol{u}\| \tag{33}$$

$$\leq (1 - \eta \mu)^H \|\boldsymbol{w} - \boldsymbol{u}\|. \tag{34}$$

As a result, we have

$$\left\| \boldsymbol{w}^{(H)} - \boldsymbol{u}^{(H)} \right\|^2 = \left\| \mathbb{E}_c \boldsymbol{w}_c^{(H)} - \mathbb{E}_c \boldsymbol{u}_c^{(H)} \right\|^2 \leq \mathbb{E}_c \left\| \boldsymbol{w}_c^{(H)} - \boldsymbol{u}_c^{(H)} \right\|^2 \tag{35}$$

$$\leq [(1 - \eta \mu)^H]^2 \|\boldsymbol{w} - \boldsymbol{u}\|^2. \tag{36}$$

Then, according to the definition of pseudo-gradients, one can obtain

$$\eta^2 H^2 \|\mathcal{G}(\boldsymbol{w}) - \mathcal{G}(\boldsymbol{u})\|^2 = \left\| \boldsymbol{w} - \boldsymbol{u} - \boldsymbol{w}^{(H)} + \boldsymbol{u}^{(H)} \right\|^2 \tag{37}$$

$$= \|\boldsymbol{w} - \boldsymbol{u}\|^2 + \left\| \boldsymbol{w}^{(H)} - \boldsymbol{u}^{(H)} \right\|^2 - 2 \left\langle \boldsymbol{w} - \boldsymbol{u}, \boldsymbol{w}^{(H)} - \boldsymbol{u}^{(H)} \right\rangle \tag{38}$$

$$\leq [1 + (1 - \eta \mu)^H] \|\boldsymbol{w} - \boldsymbol{u}\|^2 - 2 \left\langle \boldsymbol{w} - \boldsymbol{u}, \boldsymbol{w}^{(H)} - \boldsymbol{u}^{(H)} \right\rangle$$
$$- (1 - \eta \mu)^H \left[ 1 - (1 - \eta \mu)^H \right] \|\boldsymbol{w} - \boldsymbol{u}\|^2 \tag{39}$$

$$= [1 + (1 - \eta \mu)^H] \left[ \|\boldsymbol{w} - \boldsymbol{u}\|^2 - \left\langle \boldsymbol{w} - \boldsymbol{u}, \boldsymbol{w}^{(H)} - \boldsymbol{u}^{(H)} \right\rangle \right]$$
$$- (1 - (1 - \eta \mu)^H) \left\langle \boldsymbol{w} - \boldsymbol{u}, \boldsymbol{w}^{(H)} - \boldsymbol{u}^{(H)} \right\rangle$$
$$- (1 - \eta \mu)^H \left[ 1 - (1 - \eta \mu)^H \right] \|\boldsymbol{w} - \boldsymbol{u}\|^2 \tag{40}$$

$$= [1 + (1 - \eta \mu)^H] \left[ \|\boldsymbol{w} - \boldsymbol{u}\|^2 - \left\langle \boldsymbol{w} - \boldsymbol{u}, \boldsymbol{w}^{(H)} - \boldsymbol{u}^{(H)} \right\rangle \right]$$
$$+ (1 - (1 - \eta \mu)^H) \left[ \|\boldsymbol{w} - \boldsymbol{u}\|^2 - \left\langle \boldsymbol{w} - \boldsymbol{u}, \boldsymbol{w}^{(H)} - \boldsymbol{u}^{(H)} \right\rangle \right]$$
$$- \left[ 1 + (1 - \eta \mu)^H \right] \left[ 1 - (1 - \eta \mu)^H \right] \|\boldsymbol{w} - \boldsymbol{u}\|^2 \tag{41}$$

Note that $\eta H \langle \boldsymbol{w} - \boldsymbol{u}, \mathcal{G}(\boldsymbol{w}) - \mathcal{G}(\boldsymbol{u}) \rangle = \|\boldsymbol{w} - \boldsymbol{u}\|^2 - \langle \boldsymbol{w} - \boldsymbol{u}, \boldsymbol{w}^{(H)} - \boldsymbol{u}^{(H)} \rangle$ and $\eta H \widetilde{L} = 1 + (1 - \eta\mu)^H$, $\eta H \widetilde{\mu} = 1 - (1 - \eta\mu)^H$, we have

$$\|\mathcal{G}(\boldsymbol{w}) - \mathcal{G}(\boldsymbol{u})\|^2 \leq (\widetilde{L} + \widetilde{\mu}) \langle \boldsymbol{w} - \boldsymbol{u}, \mathcal{G}(\boldsymbol{w}) - \mathcal{G}(\boldsymbol{u}) \rangle - \widetilde{L}\widetilde{\mu} \|\boldsymbol{w} - \boldsymbol{u}\|^2 \tag{42}$$

$$= \widetilde{L} \langle \boldsymbol{w} - \boldsymbol{u}, \mathcal{G}(\boldsymbol{w}) - \mathcal{G}(\boldsymbol{u}) \rangle$$

$$- \widetilde{\mu} \left[ \widetilde{L} \|\boldsymbol{w} - \boldsymbol{u}\|^2 - \langle \boldsymbol{w} - \boldsymbol{u}, \mathcal{G}(\boldsymbol{w}) - \mathcal{G}(\boldsymbol{u}) \rangle \right] \tag{43}$$

$$\leq \widetilde{L} \langle \boldsymbol{w} - \boldsymbol{u}, \mathcal{G}(\boldsymbol{w}) - \mathcal{G}(\boldsymbol{u}) \rangle \tag{44}$$

where the last inequality is due to the smoothness of the pseudo-gradient (20). $\qquad\square$

### B.2 Main Proof

In the analysis below, we first analyze the training progress within one round. Suppose the current global model is $\boldsymbol{w}$ and the next round's global model is $\boldsymbol{w}^+$. Without otherwise stated, the expectation $\mathbb{E}$ and variance Var are conditioned on the current global model $\boldsymbol{w}$. For the ease of writing, we define effective learning rate $\widetilde{\alpha} = \alpha\eta H$.

First, according to the update rule of FEDAVG, we have

$$\mathbb{E}\left\| \boldsymbol{w}^+ - \boldsymbol{w}^* \right\|^2 = \mathbb{E}\left\| \boldsymbol{w} - \widetilde{\alpha}\hat{\mathcal{G}}(\boldsymbol{w}) - \boldsymbol{w}^* \right\|^2 \tag{45}$$

$$= \left\| \boldsymbol{w} - \widetilde{\alpha}\mathbb{E}[\hat{\mathcal{G}}(\boldsymbol{w})] - \boldsymbol{w}^* \right\|^2 + \widetilde{\alpha}^2 \mathrm{Var}[\hat{\mathcal{G}}(\boldsymbol{w})] \tag{46}$$

$$= \|\boldsymbol{w} - \boldsymbol{w}^*\|^2 + \widetilde{\alpha}^2 \left\| \mathbb{E}[\hat{\mathcal{G}}(\boldsymbol{w})] \right\|^2 - 2\widetilde{\alpha} \left\langle \boldsymbol{w} - \boldsymbol{w}^*, \mathbb{E}[\hat{\mathcal{G}}(\boldsymbol{w})] \right\rangle + \widetilde{\alpha}^2 \mathrm{Var}[\hat{\mathcal{G}}(\boldsymbol{w})] \tag{47}$$

Also, note that $\mathbb{E}[\hat{\mathcal{G}}(\boldsymbol{w})] = \mathcal{G}(\boldsymbol{w}) - \mathcal{G}(\boldsymbol{w}^*) + \mathcal{G}(\boldsymbol{w}^*) + \delta(\boldsymbol{w})$. So one can obtain

$$\mathbb{E}\left\| \boldsymbol{w}^+ - \boldsymbol{w}^* \right\|^2 \leq \|\boldsymbol{w} - \boldsymbol{w}^*\|^2 + 2\widetilde{\alpha}^2 \|\mathcal{G}(\boldsymbol{w}) - \mathcal{G}(\boldsymbol{w}^*)\|^2 - 2\widetilde{\alpha} \langle \boldsymbol{w} - \boldsymbol{w}^*, \mathcal{G}(\boldsymbol{w}) - \mathcal{G}(\boldsymbol{w}^*) \rangle$$
$$+ 2\widetilde{\alpha}^2 \|\mathcal{G}(\boldsymbol{w}^*) + \delta(\boldsymbol{w})\|^2 - 2\widetilde{\alpha} \langle \boldsymbol{w} - \boldsymbol{w}^*, \mathcal{G}(\boldsymbol{w}^*) + \delta(\boldsymbol{w}) \rangle + \widetilde{\alpha}^2 \mathrm{Var}[\hat{\mathcal{G}}(\boldsymbol{w})] \tag{48}$$
$$\leq (1 - \widetilde{\alpha}\widetilde{\mu}) \|\boldsymbol{w} - \boldsymbol{w}^*\|^2 + 2\widetilde{\alpha}^2 \|\mathcal{G}(\boldsymbol{w}) - \mathcal{G}(\boldsymbol{w}^*)\|^2 - \widetilde{\alpha} \langle \boldsymbol{w} - \boldsymbol{w}^*, \mathcal{G}(\boldsymbol{w}) - \mathcal{G}(\boldsymbol{w}^*) \rangle$$
$$+ 2\widetilde{\alpha}^2 \|\mathcal{G}(\boldsymbol{w}^*) + \delta(\boldsymbol{w})\|^2 - 2\widetilde{\alpha} \langle \boldsymbol{w} - \boldsymbol{w}^*, \mathcal{G}(\boldsymbol{w}^*) + \delta(\boldsymbol{w}) \rangle + \widetilde{\alpha}^2 \mathrm{Var}[\hat{\mathcal{G}}(\boldsymbol{w})] \tag{49}$$

where the first inequality uses the fact $\|a + b\|^2 \leq 2\|a\|^2 + 2\|b\|^2$, and the second inequality comes from the strongly-convexity of the pseudo-gradient. Now let us check the value of the following terms:

$$T_1 = 2\widetilde{\alpha} \|\mathcal{G}(\boldsymbol{w}) - \mathcal{G}(\boldsymbol{w}^*)\|^2 - \langle \boldsymbol{w} - \boldsymbol{w}^*, \mathcal{G}(\boldsymbol{w}) - \mathcal{G}(\boldsymbol{w}^*) \rangle - 2 \langle \boldsymbol{w} - \boldsymbol{w}^*, \mathcal{G}(\boldsymbol{w}^*) + \delta(\boldsymbol{w}) \rangle. \tag{50}$$

According to the co-coercivity of the pseudo-gradient, we have

$$T_1 \leq \left[ 2\widetilde{\alpha}\widetilde{L} - 1 \right] \langle \boldsymbol{w} - \boldsymbol{w}^*, \mathcal{G}(\boldsymbol{w}) - \mathcal{G}(\boldsymbol{w}^*) \rangle - 2 \langle \boldsymbol{w} - \boldsymbol{w}^*, \mathcal{G}(\boldsymbol{w}^*) + \delta(\boldsymbol{w}) \rangle \tag{51}$$

$$\leq \left[ 2\widetilde{\alpha}\widetilde{L} - 1 \right] \langle \boldsymbol{w} - \boldsymbol{w}^*, \mathcal{G}(\boldsymbol{w}) - \mathcal{G}(\boldsymbol{w}^*) \rangle + \epsilon \|\boldsymbol{w} - \boldsymbol{w}^*\|^2 + \frac{1}{\epsilon} \|\mathcal{G}(\boldsymbol{w}^*) + \delta(\boldsymbol{w})\|^2 \tag{52}$$

where the last inequality comes from Young's inequality. When $\widetilde{\alpha}\widetilde{\mu} \leq \widetilde{\alpha}\widetilde{L} \leq 1/4$, we have

$$T_1 \leq -\frac{\widetilde{\mu}}{2} \|\boldsymbol{w} - \boldsymbol{w}^*\|^2 + \epsilon \|\boldsymbol{w} - \boldsymbol{w}^*\|^2 + \frac{1}{\epsilon} \|\mathcal{G}(\boldsymbol{w}^*) + \delta(\boldsymbol{w})\|^2 \leq \frac{2 \|\mathcal{G}(\boldsymbol{w}^*) + \delta(\boldsymbol{w})\|^2}{\widetilde{\mu}} \tag{53}$$

where the last inequality is obtained by setting $\epsilon = \widetilde{\mu}/2$. Then, substituting (53) into (49) and noting that $\widetilde{\alpha}\widetilde{\mu} \leq \widetilde{\alpha}\widetilde{L} \leq 1/4$ (that is, $\widetilde{\alpha} \leq 1/(4\widetilde{\mu})$),

$$\mathbb{E}\left\|\boldsymbol{w}^+ - \boldsymbol{w}^*\right\|^2 \leq (1 - \widetilde{\alpha}\widetilde{\mu})\left\|\boldsymbol{w} - \boldsymbol{w}^*\right\|^2 + \left(2\widetilde{\alpha}^2 + \frac{2\widetilde{\alpha}}{\widetilde{\mu}}\right)\|\mathcal{G}(\boldsymbol{w}^*) + \delta(\boldsymbol{w})\|^2 + \widetilde{\alpha}^2\mathrm{Var}[\hat{\mathcal{G}}(\boldsymbol{w})] \tag{54}$$

$$\leq (1 - \widetilde{\alpha}\widetilde{\mu})\left\|\boldsymbol{w} - \boldsymbol{w}^*\right\|^2 + \frac{5\widetilde{\alpha}}{2\widetilde{\mu}}\|\mathcal{G}(\boldsymbol{w}^*) + \delta(\boldsymbol{w})\|^2 + \widetilde{\alpha}^2\mathrm{Var}[\hat{\mathcal{G}}(\boldsymbol{w})] \tag{55}$$

$$\leq (1 - \widetilde{\alpha}\widetilde{\mu})\left\|\boldsymbol{w} - \boldsymbol{w}^*\right\|^2 + \widetilde{\alpha}^2\mathrm{Var}[\hat{\mathcal{G}}(\boldsymbol{w})] + \frac{5\widetilde{\alpha}}{\widetilde{\mu}}\|\delta(\boldsymbol{w})\|^2 + \frac{5\widetilde{\alpha}}{\widetilde{\mu}}\|\mathcal{G}(\boldsymbol{w}^*)\|^2 \tag{56}$$

$$\leq (1 - \widetilde{\alpha}\widetilde{\mu})\left\|\boldsymbol{w} - \boldsymbol{w}^*\right\|^2 + \widetilde{\alpha}^2 \max_{\boldsymbol{w}}\mathrm{Var}[\hat{\mathcal{G}}(\boldsymbol{w})]$$
$$+ \frac{5\widetilde{\alpha}}{\widetilde{\mu}}\max_{\boldsymbol{w}}\|\delta(\boldsymbol{w})\|^2 + \frac{5\widetilde{\alpha}}{\widetilde{\mu}}\|\mathcal{G}(\boldsymbol{w}^*)\|^2 \tag{57}$$

After total $T$ communication rounds and taking the total expectation, we end up with

$$\mathbb{E}\left\|\boldsymbol{w}^{(T)} - \boldsymbol{w}^*\right\|^2 \leq (1 - \widetilde{\alpha}\widetilde{\mu})^T\left\|\boldsymbol{w}^{(0)} - \boldsymbol{w}^*\right\|^2 + \frac{\widetilde{\alpha}}{\widetilde{\mu}}\max_{\boldsymbol{w}}\mathrm{Var}[\hat{\mathcal{G}}(\boldsymbol{w})]$$
$$+ \frac{5}{\widetilde{\mu}^2}\max_{\boldsymbol{w}}\mathbb{E}_c\|\delta_c(\boldsymbol{w})\|^2 + \frac{5\rho^2}{\widetilde{\mu}^2}. \tag{58}$$

When $\eta H\mu \leq 1$, one can easily validate that

$$\widetilde{\mu} = \frac{1 - (1 - \eta\mu)^H}{\eta H} \geq \frac{\mu}{2}. \tag{59}$$

So it follows that

$$\mathbb{E}\left\|\boldsymbol{w}^{(T)} - \boldsymbol{w}^*\right\|^2 \leq (1 - \frac{1}{2}\alpha\eta H\mu)^T\left\|\boldsymbol{w}^{(0)} - \boldsymbol{w}^*\right\|^2 + \frac{2\alpha\eta H}{\mu}\max_{\boldsymbol{w}}\mathrm{Var}[\hat{\mathcal{G}}(\boldsymbol{w})]$$
$$+ \frac{20}{\mu^2}\max_{\boldsymbol{w}}\mathbb{E}_c\|\delta_c(\boldsymbol{w})\|^2 + \frac{20\rho^2}{\mu^2}. \tag{60}$$

At last, in order to satisfy $\widetilde{\alpha}\widetilde{L} \leq 1/4$, one can set $\alpha \leq 1/8$, such that

$$\widetilde{\alpha}\widetilde{L} = \alpha\eta H \cdot \frac{1 + (1 - \eta\mu)^H}{\eta H} = \alpha(1 + (1 - \eta\mu)^H) \leq 2\alpha \leq \frac{1}{4}. \tag{61}$$

## C   Bound on Iterate Bias

In this section, we will provide an upper bound for the iterate bias (11). According to the local update rules, we have

$$\left\|\boldsymbol{w}_{c,\mathrm{GD}}^{(H)} - \mathbb{E}[\boldsymbol{w}_c^{(H)}]\right\| = \left\|\boldsymbol{w}_{c,\mathrm{GD}}^{(H-1)} - \mathbb{E}[\boldsymbol{w}_c^{(H-1)}] - \eta\nabla F_c(\boldsymbol{w}_{c,\mathrm{GD}}^{(H-1)}) + \eta\mathbb{E}[\nabla F_c(\boldsymbol{w}_c^{(H-1)})]\right\| \tag{62}$$

$$\leq \left\|\boldsymbol{w}_{c,\mathrm{GD}}^{(H-1)} - \mathbb{E}[\boldsymbol{w}_c^{(H-1)}] - \eta\nabla F_c(\boldsymbol{w}_{c,\mathrm{GD}}^{(H-1)}) + \eta\nabla F_c(\mathbb{E}[\boldsymbol{w}_c^{(H-1)}])\right\|$$
$$+ \eta\left\|\mathbb{E}[\nabla F_c(\boldsymbol{w}_c^{(H-1)})] - \nabla F_c(\mathbb{E}[\boldsymbol{w}_c^{(H-1)}])\right\| \tag{63}$$

$$\leq (1 - \eta\mu)\left\|\boldsymbol{w}_{c,\mathrm{GD}}^{(H-1)} - \mathbb{E}[\boldsymbol{w}_c^{(H-1)}]\right\|$$
$$+ \eta\left\|\mathbb{E}[\nabla F_c(\boldsymbol{w}_c^{(H-1)})] - \nabla F_c(\mathbb{E}[\boldsymbol{w}_c^{(H-1)}])\right\| \tag{64}$$

For the second term, we have

$$\left\|\mathbb{E}[\nabla F_c(\boldsymbol{w}_c^{(H-1)})] - \nabla F_c(\mathbb{E}[\boldsymbol{w}_c^{(H-1)}])\right\|^2 \leq \mathbb{E}\left\|\nabla F_c(\boldsymbol{w}_c^{(H-1)}) - \nabla F_c(\mathbb{E}[\boldsymbol{w}_c^{(H-1)}])\right\|^2 \tag{65}$$

$$\leq L^2\mathbb{E}\left\|\boldsymbol{w}_c^{H-1} - \mathbb{E}[\boldsymbol{w}_c^{(H-1)}]\right\|^2 \tag{66}$$

$$\leq 2\eta^2 L^2\sigma^2(H-1) \tag{67}$$

where the last inequality comes from previous works Khaled et al. (2020); Glasgow et al. (2021). As a result, one can obtain

$$\left\| \boldsymbol{w}_{c,\text{GD}}^{(H)} - \mathbb{E}[\boldsymbol{w}_c^{(H)}] \right\| \leq (1 - \eta\mu) \left\| \boldsymbol{w}_{c,\text{GD}}^{(H-1)} - \mathbb{E}[\boldsymbol{w}_c^{(H-1)}] \right\| + \sqrt{2}\eta^2 L\sigma(H-1)^{\frac{1}{2}} \tag{68}$$

$$\leq \sqrt{2}\eta^2 L\sigma \sum_{h=0}^{H-1} (1 - \eta\mu)^{H-1-h} h^{\frac{1}{2}} \tag{69}$$

$$\leq \sqrt{2}\eta^2 L\sigma \left[ \sum_{h=0}^{H-1} (1 - \eta\mu)^{2(H-1-h)} \right]^{\frac{1}{2}} \left[ \sum_{h=0}^{H-1} h \right]^{\frac{1}{2}} \tag{70}$$

$$\leq \sqrt{2}\eta^2 L\sigma \left[ \sum_{h=0}^{H-1} (1 - \eta\mu)^{H-1-h} \right]^{\frac{1}{2}} \left[ \sum_{h=0}^{H-1} h \right]^{\frac{1}{2}} \tag{71}$$

$$= \left[ \frac{1 - (1-\eta\mu)^H}{\eta\mu H} \right]^{\frac{1}{2}} \eta^2 L\sigma H (H-1)^{\frac{1}{2}}. \tag{72}$$

According to the definition of $\delta(\boldsymbol{w})$, we obtain

$$\mathbb{E}_c \left\| \delta_c(\boldsymbol{w}) \right\|^2 \leq \mathbb{E}_c \left\| \boldsymbol{w}_{c,\text{GD}}^{(H)} - \mathbb{E}[\boldsymbol{w}_c^{(H)}] \right\|^2 \leq \frac{\widetilde{\mu}\eta^2 L^2\sigma^2(H-1)}{\mu} \leq \eta^2 L^2\sigma^2(H-1) \tag{73}$$

where the last inequality comes from the fact that $\widetilde{\mu} \leq \mu$.

## D   Proof of Corollary 1

### D.1   Deterministic Setting

When clients perform local GD in each round, there is no stochastic noise. So the error upper bound (9) can be simplified as follows

$$\left\| \boldsymbol{w}^{(T)} - \boldsymbol{w}^* \right\|^2 \leq (1 - \tfrac{1}{2}\alpha\eta H\mu)^T \left\| \boldsymbol{w}^{(0)} - \boldsymbol{w}^* \right\|^2 + \frac{20\rho^2}{\mu^2}. \tag{74}$$

If $H\mu \geq L$, then the maximal learning rate is $\eta = 1/H\mu$. When $\alpha = 1/8$, the upper bound becomes

$$\left\| \boldsymbol{w}^{(T)} - \boldsymbol{w}^* \right\|^2 \leq \left( 1 - \frac{1}{16} \right)^T \left\| \boldsymbol{w}^{(0)} - \boldsymbol{w}^* \right\|^2 + \frac{20\rho^2}{\mu^2} \tag{75}$$

$$\leq \exp\left( -\frac{T}{16} \right) \left\| \boldsymbol{w}^{(0)} - \boldsymbol{w}^* \right\|^2 + \frac{20\rho^2}{\mu^2}. \tag{76}$$

If $H\mu \leq L$, then the maximal learning rate is $\eta = 1/L$. When $\alpha = 1/8$, the upper bound becomes

$$\left\| \boldsymbol{w}^{(T)} - \boldsymbol{w}^* \right\|^2 \leq \left( 1 - \frac{H\mu}{16L} \right)^T \left\| \boldsymbol{w}^{(0)} - \boldsymbol{w}^* \right\|^2 + \frac{20\rho^2}{\mu^2} \tag{77}$$

$$\leq \exp\left( -\frac{\mu H T}{16L} \right) \left\| \boldsymbol{w}^{(0)} - \boldsymbol{w}^* \right\|^2 + \frac{20\rho^2}{\mu^2}. \tag{78}$$

We can summarize the above two bounds as follows:

$$\left\| \boldsymbol{w}^{(T)} - \boldsymbol{w}^* \right\|^2 \leq \exp\left( -\frac{T}{16\kappa} \min\{\kappa, H\} \right) \left\| \boldsymbol{w}^{(0)} - \boldsymbol{w}^* \right\|^2 + \frac{20\rho^2}{\mu^2} \tag{79}$$

$$= \mathcal{O}\left( \exp\left( -\frac{T}{16\kappa} \min\{\kappa, H\} \right) + \rho^2 \right). \tag{80}$$

## D.2 Stochastic Setting

Substituting the upper bounds for $\mathrm{Var}[\mathcal{G}(\boldsymbol{w})]$ and $\delta(\boldsymbol{w})$ into (57) and setting $\alpha = 1/8$,

$$\mathbb{E}\left\|\boldsymbol{w}^{(t+1)} - \boldsymbol{w}^*\right\|^2 \leq (1 - \widetilde{\alpha}\widetilde{\mu})\left\|\boldsymbol{w}^{(t)} - \boldsymbol{w}^*\right\|^2 + \widetilde{\alpha}^2 \frac{2\sigma^2}{MH} + \frac{5\widetilde{\alpha}^3}{\widetilde{\mu}}\frac{\sigma^2 L^2(H-1)}{\alpha^2 H^2} + \frac{5\widetilde{\alpha}}{\widetilde{\mu}}\left\|\mathcal{G}(\boldsymbol{w}^*)\right\|^2 \tag{81}$$

$$\leq (1 - \widetilde{\alpha}\widetilde{\mu})\left\|\boldsymbol{w}^{(t)} - \boldsymbol{w}^*\right\|^2 + \widetilde{\alpha}^2 \frac{2\sigma^2}{MH} + \widetilde{\alpha}^3 \frac{320\sigma^2 L^2}{\widetilde{\mu}H} + \widetilde{\alpha}\frac{5\rho^2}{\widetilde{\mu}}. \tag{82}$$

After minor rearrangement, we can get

$$\mathbb{E}\left\|\boldsymbol{w}^{(t+1)} - \boldsymbol{w}^*\right\|^2 - \frac{5\rho^2}{\widetilde{\mu}^2} \leq (1 - \widetilde{\alpha}\widetilde{\mu})\left[\left\|\boldsymbol{w}^{(t)} - \boldsymbol{w}^*\right\|^2 - \frac{5\rho^2}{\widetilde{\mu}^2}\right] + \widetilde{\alpha}^2 \frac{2\sigma^2}{MH} + \widetilde{\alpha}^3 \frac{320\sigma^2 L^2}{\widetilde{\mu}H}. \tag{83}$$

After total $T$ communication rounds,

$$\mathbb{E}\left\|\boldsymbol{w}^{(t+1)} - \boldsymbol{w}^*\right\|^2 - \frac{5\rho^2}{\widetilde{\mu}^2} \leq (1 - \widetilde{\alpha}\widetilde{\mu})^T \underbrace{\left[\left\|\boldsymbol{w}^{(0)} - \boldsymbol{w}^*\right\|^2 - \frac{5\rho^2}{\widetilde{\mu}^2}\right]}_{r_0} + \frac{2\widetilde{\alpha}\sigma^2}{\widetilde{\mu}MH} + \frac{320\widetilde{\alpha}^2\sigma^2 L^2}{\widetilde{\mu}^2 H}. \tag{84}$$

If we set $\widetilde{\alpha} = \frac{\nu}{\mu T}$, where $\nu = 2\ln(\max\{r_0\mu^2 MHT/(8\sigma^2), r_0\mu^4 HT^2/(1280L^2\sigma^2)\})$, then it follows that

$$\mathbb{E}\left\|\boldsymbol{w}^{(t+1)} - \boldsymbol{w}^*\right\|^2 - \frac{20\rho^2}{\mu^2} \leq \frac{8\sigma^2\nu}{\mu^2 MHT} + \frac{1280\sigma^2 L^2\nu}{\mu^4 HT^2} + \exp\left(-\frac{\nu}{2}\right)r_0 \tag{85}$$

$$\leq \frac{8\sigma^2(\nu+1)}{\mu^2 MHT} + \frac{1280\sigma^2 L^2(\nu+1)}{\mu^4 HT^2} \tag{86}$$

$$= \widetilde{\mathcal{O}}\left(\frac{\sigma^2}{MHT} + \frac{\sigma^2}{HT^2}\right) \tag{87}$$

where $\widetilde{\mathcal{O}}$ hides logarithmic factors.

# E Extensions to General Convex Functions

In this section, we are going to extend Theorem 2 to general convex settings for deterministic FEDAVG. This extension can help to show that our conclusion "FedAvg can have identical convergence rate in homogeneous and heterogeneous settings" is not only constrained to the strongly convex settings.

When the average client drift at optimum is zero and there is no stochastic noise, we have

$$\left\|\boldsymbol{w}^+ - \boldsymbol{w}^*\right\|^2 = \left\|\boldsymbol{w} - \boldsymbol{w}^* - \alpha\mathcal{G}(\boldsymbol{w})\right\|^2 \tag{88}$$

$$= \left\|\boldsymbol{w} - \boldsymbol{w}^*\right\|^2 - 2\alpha\left\langle\boldsymbol{w} - \boldsymbol{w}^*, \mathcal{G}(\boldsymbol{w}) - \mathcal{G}(\boldsymbol{w}^*)\right\rangle + \alpha^2\left\|\mathcal{G}(\boldsymbol{w}) - \mathcal{G}(\boldsymbol{w}^*)\right\|^2 \tag{89}$$

$$\leq (1 - \alpha\widetilde{\mu})\left\|\boldsymbol{w} - \boldsymbol{w}^*\right\|^2 - \frac{\alpha}{\widetilde{L}}(1 - \alpha\widetilde{L})\left\|\mathcal{G}(\boldsymbol{w}) - \mathcal{G}(\boldsymbol{w}^*)\right\|^2 \tag{90}$$

$$\leq (1 - \alpha\widetilde{\mu})\left\|\boldsymbol{w} - \boldsymbol{w}^*\right\|^2 - \frac{\alpha}{2\widetilde{L}}\left\|\mathcal{G}(\boldsymbol{w}) - \mathcal{G}(\boldsymbol{w}^*)\right\|^2 \tag{91}$$

where the last inequality is due to $\alpha\widetilde{L} \leq 1/2$. In the general convex setting, we have $\mu = 0$. According to the definitions of $\widetilde{\mu}, \widetilde{L}$ in Lemma 2, it follows that $\widetilde{\mu} = 0$ and $\widetilde{L} = 2/\eta H$. Substituting these values into (91), we obtain

$$\left\|\boldsymbol{w}^+ - \boldsymbol{w}^*\right\|^2 \leq \left\|\boldsymbol{w} - \boldsymbol{w}^*\right\|^2 - \frac{\alpha\eta H}{4}\left\|\mathcal{G}(\boldsymbol{w}) - \mathcal{G}(\boldsymbol{w}^*)\right\|^2. \tag{92}$$

After minor rearrangement, we have

$$\left\|\mathcal{G}(\boldsymbol{w}) - \mathcal{G}(\boldsymbol{w}^*)\right\|^2 \leq \frac{4}{\alpha\eta H}\left[\left\|\boldsymbol{w} - \boldsymbol{w}^*\right\|^2 - \left\|\boldsymbol{w}^+ - \boldsymbol{w}^*\right\|^2\right]. \tag{93}$$

Taking the average from $t = 0$ to $t = T - 1$,

$$\frac{1}{T} \sum_{t=0}^{T-1} \left\| \mathcal{G}(\boldsymbol{w}^{(t)}) - \mathcal{G}(\boldsymbol{w}^*) \right\|^2 \leq \frac{4 \left\| \boldsymbol{w}^{(0)} - \boldsymbol{w}^* \right\|^2}{\alpha \eta H T} \tag{94}$$

If we set $\alpha = 1/4, \eta = 1/L$, then

$$\frac{1}{T} \sum_{t=0}^{T-1} \left\| \mathcal{G}(\boldsymbol{w}^{(t)}) - \mathcal{G}(\boldsymbol{w}^*) \right\|^2 \leq \frac{16L \left\| \boldsymbol{w}^{(0)} - \boldsymbol{w}^* \right\|^2}{H T}. \tag{95}$$

The above rate is the same as GD for general convex functions and data heterogeneity does not have negative impacts, as $\rho = \|\mathcal{G}(\boldsymbol{w})\| = 0$.

# F   Proof of Theorem 3

*Proof.* According to the local update rule, we have

$$\boldsymbol{w}_c^{(t,h+1)} = \boldsymbol{w}_c^{(t,h)} - \eta \nabla F_c(\boldsymbol{w}_c^{(t,h)}) \tag{96}$$

$$= \boldsymbol{w}_c^{(t,h)} - \eta \left[ \boldsymbol{A}_c(\boldsymbol{w}_c^{(t,h)} - \boldsymbol{w}^*) - \boldsymbol{b}_c \right] \tag{97}$$

$$= (\boldsymbol{I} - \eta \boldsymbol{A}_c) \boldsymbol{w}_c^{(t,h)} + \eta (\boldsymbol{A}_c \boldsymbol{w}^* + \boldsymbol{b}_c). \tag{98}$$

Subtracting $\boldsymbol{w}_c^* = \boldsymbol{w}^* + \boldsymbol{A}_c^{-1} \boldsymbol{b}_c$ on both sides, it follows that

$$\boldsymbol{w}_c^{(t,h+1)} - \boldsymbol{w}_c^* = (\boldsymbol{I} - \eta \boldsymbol{A}_c) \left( \boldsymbol{w}_c^{(t,h)} - \boldsymbol{w}_c^* \right) \tag{99}$$

$$= (\boldsymbol{I} - \eta \boldsymbol{A}_c)^{h+1} \left( \boldsymbol{w}^{(t)} - \boldsymbol{w}_c^* \right). \tag{100}$$

Setting $h = H$, we have $\boldsymbol{w}_c^{(t,H)} = (\boldsymbol{I} - \eta \boldsymbol{A}_c)^H (\boldsymbol{w}^{(t)} - \boldsymbol{w}_c^*) + \boldsymbol{w}_c^*$. Recall the definition of pseudo-gradient (6), we get

$$\mathcal{G}_c(\boldsymbol{w}^{(t)}) = \frac{1}{\eta H} (\boldsymbol{w}^{(t)} - \boldsymbol{w}^{(t,H)}) \tag{101}$$

$$= \frac{1}{\eta H} \left[ \boldsymbol{I} - (\boldsymbol{I} - \eta \boldsymbol{A}_c)^H \right] (\boldsymbol{w}^{(t)} - \boldsymbol{w}_c^*). \tag{102}$$

According to the global update rule of FEDAVG, one can obtain that

$$\boldsymbol{w}^{(t+1)} = \boldsymbol{w}^{(t)} - \alpha \eta H \mathbb{E}_c \mathcal{G}_c(\boldsymbol{w}^{(t)}) \tag{103}$$

$$= \boldsymbol{w}^{(t)} - \alpha \mathbb{E}_c \left[ (\boldsymbol{I} - (\boldsymbol{I} - \eta \boldsymbol{A}_c)^H) \left( \boldsymbol{w}^{(t)} - \boldsymbol{w}_c^* \right) \right] \tag{104}$$

$$= \boldsymbol{w}^{(t)} - \alpha \mathbb{E}_c \left[ (\boldsymbol{I} - (\boldsymbol{I} - \eta \boldsymbol{A}_c)^H) \left( \boldsymbol{w}^{(t)} - \boldsymbol{w}^* \right) \right]$$
$$- \alpha \mathbb{E}_c \left[ (\boldsymbol{I} - (\boldsymbol{I} - \eta \boldsymbol{A}_c)^H) (\boldsymbol{w}^* - \boldsymbol{w}_c^*) \right]. \tag{105}$$

Subtracting $\boldsymbol{w}^*$ on both sides and setting $\alpha = 1$, we have

$$\boldsymbol{w}^{(t+1)} - \boldsymbol{w}^* = \mathbb{E}_c \left[ (\boldsymbol{I} - \eta \boldsymbol{A}_c)^H \right] \left( \boldsymbol{w}^{(t)} - \boldsymbol{w}^* \right) - \mathbb{E}_c \left[ \underbrace{(\boldsymbol{I} - (\boldsymbol{I} - \eta \boldsymbol{A}_c)^H) (\boldsymbol{w}^* - \boldsymbol{w}_c^*)}_{\eta H \mathcal{G}_c(\boldsymbol{w}^*)} \right] \tag{106}$$

$$= \mathbb{E}_c \left[ (\boldsymbol{I} - \eta \boldsymbol{A}_c)^H \right] \left( \boldsymbol{w}^{(t)} - \boldsymbol{w}^* \right) - \eta H \mathcal{G}(\boldsymbol{w}^*) \tag{107}$$

where $\mathcal{G}(\boldsymbol{w}^*) = \mathbb{E}_c \mathcal{G}_c(\boldsymbol{w}^*)$.

Now, we prove that $\mathcal{G}(\boldsymbol{w}^*) = 0$ almost surely as $M \to \infty$ on this synthetic problem. According to the definition of $\boldsymbol{A}_c, \boldsymbol{b}_c$, we obtain that

$$\mathcal{G}(\boldsymbol{w}^*) = \mathbb{E}_c \left[ (\boldsymbol{I} - (\boldsymbol{I} - \eta \boldsymbol{A}_c)^H) \boldsymbol{A}_c^{-1} \boldsymbol{b}_c \right] \tag{108}$$

$$= \mathbb{E}_c \left[ \underbrace{(\boldsymbol{I} - (\boldsymbol{I} - \eta \boldsymbol{A}_c)^H) \boldsymbol{A}_c^{-1} \frac{1}{n} \sum_{i=1}^n \boldsymbol{x}_{c,i} \epsilon_{c,i}}_{\xi_c} \right]. \tag{109}$$

Since the noise $\epsilon_{c,\cdot}$ are independent of $\boldsymbol{x}_{c,\cdot}$, $\xi_c$ is a zero-mean random variable that depends on client $c$. Since we have assumed that all $\|\boldsymbol{x}_{c,i}\|$ and $\epsilon_{c,i}$ have bounded variance, we know that $\mathbb{E}[\xi_c^2] < \infty$. Since we have a uniform weighting on the $M$ clients, it follows $\mathbb{E}_c[\xi_c] = O(1/\sqrt{M})$ with probability $1 - o_M(1)$, and as $M \to \infty$, we have $\mathbb{E}_c[\xi_c] \to 0$ almost surely.

Hence, from (107), we conclude that

$$\boldsymbol{w}^{(t+1)} - \boldsymbol{w}^* = \left[ \mathbb{E}_c \left[ (\boldsymbol{I} - \eta \boldsymbol{A}_c)^H \right] \right]^{t+1} (\boldsymbol{w}^{(0)} - \boldsymbol{w}^*), \tag{110}$$

almost surely as $M \to \infty$, which proves the desired result.

$\square$

