# OpenReview forum: "On the Unreasonable Effectiveness of Federated Averaging with Heterogeneous Data"
_TMLR — Accepted by TMLR_

### Review · Reviewer_TjzG · 2024-03-15

**Summary Of Contributions:**

The paper looks into an apparent mismatch between theory and practice in federated learning with FedAvg. Current theory suggests that client heterogeneity incurs a strong penalty on the convergence, whereas empirical results on at least some heterogeneous data sets are not improved by shuffling, indicating that heterogeneity is not always a practical problem. The paper proposes a simple condition (that the client gradients average to zero at the optimum, i.e. that FedAvg converges) and show that this leads to better convergence bounds. This is most clearly shown in a simple linear regression example.

**Audience:**

Yes

**Claims And Evidence:**

No

**Requested Changes:**

I noticed that there is only one paper listed from 2023 in the references. Perhaps there are more recent studies that could be relevant to include?

The statement that SCAFFOLD and FedAvg have roughly identical performance is supported only by one non-peer-reviewed reference (Reddi et al. 2020). I do not consider this claim to be well enough established. Most other work I have seen, seem to indicate that SCAFFOLD works quite well in many settings, including some real world applications. There are also extensions of SCAFFOLD that improve further upon this, indicating that variance reduction can yield strong improvements.

"mysterious inconsistency" I am not sure there is any mystery or inconsistency. But a gap between theory and practice, perhaps.

Is "simulated through pathological datasets" an accurate characterization of previous work? In which exact sense can you underpin that most simulation studies that are pathological?

The second bullet point in the main contribution says "...whenever FedAvg converges...we prove..there is no negative impact on the convergence...even with unbounded gradient dissimilarity." I believe I understand the gist of what you're trying to convey, but I find it a bit unclear.

"We further validate that the client consensus hypothesis is realistic." I'm uncertain about what exactly needs to be validated, given that the hypothesis revolves around the convergence of FedAvg. As far as I know, the criteria for the convergence of FedAvg are well established. I'm open to clarification if there are nuances I may have missed.

Question: Is discussing only the full device participation case is a significant limitation of the paper. Presumable the influence of data heterogeneity is significantly stronger in the partial participation case? For cases when your 'consensus hypothesis' does not hold strongly, does partial participation not influence performance?

I respectfully disagree with the characterization of classification dataset with unequal class balance across clients as 'pathological'. In many instances, such imbalances hold practical significance.

"In order to support the above conjecture..." I believe it would be beneficial to provide clearer motivation regarding the presence of a gap between theory and practice. While discovering one or more examples demonstrating natural heterogeneity where shuffling doesn't improve convergence isn't evidence that data heterogeneity isn't a practical concern in other scenarios.

I think some effort should be put into examining how often the 'consensus hypothesis' holds in relevant scenarios, or to define more clear criteria for when it holds, or to outline methods to to determine if it holds. The paper seems to suggest that this hypothesis often holds, but there is not much clear evidence to support this.

**Strengths And Weaknesses:**

Strengths
* The ideas are simple and clear, and should be simple to confirm and reproduce.
* Research questions are well defined and clearly guide the paper.
* The paper provides important intuition about when FedAvg can be expected to be effective and when not.
* The writing is accessible.

Weaknesses
* Experiments could have been carried out at a larger scale to give a better impression of how often the hypothesis holds in practice over some large set of real problems.
* The linear regression example is a clear demonstration of the case when the 'client consensus' hypothesis holds. It would be very clear if the paper also gave an equally simple example where the hypothesis does not hold. In general, the paper could benefit from providing the reader a more clear intuition to which cases the hypothesis is likely to hold.

---

> ### Author Response · Authors · 2024-04-20
> **Author response**
>
> We thank the reviewer for pointing out that our ideas are simple and clear, and providing important intuition. Below, we response 1-by-1 to the requested changes:
> 1. Thanks for the suggestion! We added a few new references from 2023 and 2024.
> 2. We added a few more peer-reviewed references, and corrected the reference of Reddi et al 2020, which is peer-reviewed and presented at ICLR 2021. In these three papers, their reported performance of SCAFFOLD is on par or slightly worse than FedAvg. So we conjecture that its performance may relate to the dataset or the overall federated settings. We rephrased the claims to avoid confusions.
> 3. Following the reviewer’s suggestion, we changed “mysterious inconsistency” to “gap”.
> 4. Thank the reviewer for pointing out this! We changed the word “pathological” to a less controversial term “artificial datasets” or “datasets with artificially high heterogeneity”.
> 5. Thanks for the suggestion! We rephrase that sentence.
> 6. We rephrased “... is realistic” to “can hold in many scenarios”.The hypothesis itself needs to be examined, not the convergence of FedAvg.
> 7. We respectfully disagree that the “full device participation case” is a big limitation. As discussed in the paragraph before section 5, we discuss how standard techniques can be used to deal with the additional variance terms introduced by client sampling. Such variance terms do not introduce additional data heterogeneity.
> 8. Thank the reviewer for pointing out this! We changed the word “pathological” to “artificial”. Hopefully this is less controversial.
> 9. We agree with the reviewer and make some changes to related claims.
> 10. As discussed in the beginning of section 5, our intuition is that the client consensus hypothesis is likely to hold when a single global model can work reasonably well for all clients. For example, when all clients have the same conditional probability of the label given feature p(y|x) while their underlying feature distribution p(x) can be drastically different. In this case, the client’s updates should not conflict with each other, as all local models tend to learn the same conditional distribution p(y|x). Following this intuition, we construct the example in section 5. Besides, it is clear that the non-IID version of CIFAR-10/100 does not satisfy the property of having the same conditional distribution. On CIFAR-10/100, it is common to divide data by labels. That is, clients have the same p(x|y) instead of p(y|x).

---

### Review · Reviewer_ZSqV · 2024-03-23

**Summary Of Contributions:**

This paper challenges common theoretical assumptions of federated learning like Lipschitz smoothness and local unbiased gradient. Instead of reusing these assumptions, it proposes client consensus hypothesis, which essentially says that data local heterogeneity will cancel out. Based on this assumption, the paper derives a new convergence bound and verifies this assumption in synthetic (linear regression) and real datasets.

**Audience:**

Yes

**Broader Impact Concerns:**

This is more like an analysis paper and I'm not sure if broader impact is required here.

**Claims And Evidence:**

No

**Requested Changes:**

Most of the changes have been mentioned above. In addition, I would request the authors to:

1. finish the theoretical proof of eq. (16) and show exactly why it converges to 0.
2. Remove the usage of "pathological" for CIFAR-100 since many people may not agree it's pathological. I think they are just different kinds of data heterogeneity.
3. Add more analysis on Assumption 4 and show in which cases (covariate shift, label shift, etc.) Assumption 4 is more likely to hold.
4. Add a table to compare existing assumptions/your assumptions and existing convergence bounds/your bounds (from Appendix A/B).
5. Compare FedAvg with the centralized setting since the latter is an upper bound. This will further demonstrate that FedAvg is not affected by data heterogeneity.

**Strengths And Weaknesses:**

Strengths:

1. Many of existing FL papers are based on similar assumptions: Lipschitz smoothness and bounded variance. This paper points out that the assumptions are too pessimistic without considering the cancelling effect of the local gradients. This approach seems novel.
2. The authors do not only derive better convergence results based on the new assumption, but also verify it in linear regression and FEMNIST and Shakespeare.

Weaknesses:
1. This paper doesn't propose a new algorithm of FL, but rather analyze the existing effectiveness of FedAvg.
2. It says that label shift split of CIFAR-100 is pathological, but it is a standard FL dataset. Whether it is pathological is still a question. For example, we could have many different users, and some users have more classes of cats than others, which is totally reasonable. Therefore, I think this paper is a bit overclaim. There are still a lot of cases when FedAvg suffer and existing FL algorithms can improve, and one should just call them pathological just because they don't satisfy the client consensus hypothesis.
3. The datasets (FEMNIST, StackOverflow) tested are still small scale.
4. The main contribution revolves around a new assumption of FL which may or may not hold in practice. The authors should add more results on how to verify/falsify the assumption in FL problems.
5. Since this is an analysis paper, I would expect more comprehensive analysis on more datasets, like classic CIFAR-10/100. Also, why does the main assumption not hold in these label shift settings?

---

> ### Author Response · Authors · 2024-04-20
> **Author response**
>
> We thank the reviewer for providing the valuable suggestions!
>
> Regarding the experiments, we wanted to note that StackOverflow is already a large-scale and a commonly used dataset, which consists of more than 300k clients and 135M data samples, as a comparison, other datasets like CIFAR-100/10 has up to hundred clients and 60k data samples, FLAIR has 50k clients and 430k data samples.
>
> Regarding other requested changes:
> 1. The proof of eq. (16) can be found at the end of Appendix in the original version.
> 2. We thank the reviewer for raising this! We changed “pathological datasets” to a less controversial term “datasets with artificially high heterogeneity”.
> 3. We discussed intuition at the beginning of section 5,the client consensus hypothesis is likely to hold when a single global model can work well for all clients. For example, when all clients have the same conditional probability of the label given feature while their underlying feature distribution can be drastically different. In this case, the client’s updates should not conflict with each other, as all local models tend to learn the same conditional distribution. Also, following this intuition, we come up with the example discussed in Section 5.1.
> 4. Thanks for the suggestion! We moved the two tables and corresponding discussions from Appendix to main text.
> 5. Thanks for the suggestion! We added some discussions at the end of Section 4.

---

### Review · Reviewer_Vef9 · 2024-04-06

**Summary Of Contributions:**

This paper observes that the FedAvg algorithm performs well on heterogeneous data in practice, but existing theory results (i.e., upper bounds) is very loose and fails to justify this good performance on heterogeneous data. This paper analyzes why the existing analysis is not tight, and proposes a new hypothesis called “client consensus hypothesis”, which basically says that the average change/updates from clients tends to zero if starting from the (global) optimum. This hypothesis is verified on the task of linear regression and on real datasets. The main result is an improved convergence bounds for FedAvg which is independent of whether or not the data is heterogeneous, assuming the client consensus hypothesis holds.

**Audience:**

Yes

**Broader Impact Concerns:**

None.

**Claims And Evidence:**

Yes

**Requested Changes:**

- In the abstract, when you mention “existing theoretical results”, maybe give some representative reference already here?

- In the abstract, the “client consensus hypothesis” may need to be rephrased, as I don’t understand this sentence from first reading it. a)“average of client updates” — maybe clarify that this is the amount of update to the gradient etc.? b) “on certain datasets” sounds weird, since any arbitrary property can be observed on “certain” dataset. So it might be good to make a stronger claim, such as “on many typical federated datasets”. Or, as a hypothesis which aims to talk about the property itself, you can remove the half sentence “on certain federated datasets”

- “Unreasonable practical effectiveness” — this “unreasonable” may not be a proper word in my opinion. It is true that it performs differently than previous theory results suggest, but this does not mean the practical effectiveness is “unreasonable”, since otherwise it would mean the experiments are not correct (which is not the case). Instead, it is the existing theory that is “unreasonable” (since it does not capture the practical cases well enough). Actually, for the title, it looks better to me if you simply remove the word “unreasonable”, so it reads “on the effectiveness of FedAvg with heterogeneous data”

- Reading your “Main Contributions”, it is mentioned that the hypothesis is satisfied as long as FedAvG converges. Does this mean as long as FedAvG converges, then it converges fast (since your theorem shows that as long as the hypothesis holds then it converges fast)? Is this message/interpretation correct? Please clarify (and possibly reflect in the paper).

- After reading Theorem 1, I’m not sure about many of your descriptions/interpretations: you seem to say previous works, including Theorem 1, says that FedAvg does *not* perform well on heterogeneous data. However, Theorem 1 is only an upper bound, so it only means FedAvg is *not known to be good* for heterogeneous data. This is much different from saying they show FedAvg is bad. Maybe you should rephrase many of your descriptions, to emphasize that a) FedAvg works well in practice and b) existing theorem fails to justify this fact (and their upper bound is very loose for heterogeneous setting).

- Did you formally define or model the data heterogeneity?

**Strengths And Weaknesses:**

Strength:

- The theory developed in this paper is elegant and looks fundamental. I think it is likely to be the correct way to explain the performance of FedAvg.

- The argument/logic is convincing, as it not only gives theoretical justification, but also verifies the hypothesis for typical real datasets.

Weakness:

- Some of the technical discussions/interpretations are not very accurate.

- The technical depth seems limited although the focus is on finding the right hypothesis. As a theory result, this might be a minor weakness.

- I’m not sure how fundamental FedAvg algorithm is, and thus this questions the motivation/significance. Alternatively, it would be great if you can justify if you can apply your technique/hypothesis to other related algorithms/problems.

- The non-convex case is not discussed, and even the hypothesis seems to only focus on the convex case (since it concerns the global optimum).

---

> ### Author Response · Authors · 2024-04-20
> **Author response**
>
> We thank the reviewer for finding the paper elegant and fundamental, and providing valuable suggestions!
>
> Regarding the requested changes, we respond to each point as follows:
> 1. We added a representative reference (Woodworth et al. 2020a) in the abstract.
> 2. In order to avoid confusion, we rephrased “client updates” to “local model updates on clients” and removed “on certain datasets”.
> 3. “Unreasonable effectiveness” in the title is a play on a classic article: https://en.wikipedia.org/wiki/The_Unreasonable_Effectiveness_of_Mathematics_in_the_Natural_Sciences and several articles use it (e.g. [1], [2], [3]). We would prefer to keep it in the title if possible. Meanwhile, we removed the usage of “unreasonable” wherever it is not sound in the main text.
> 4. We thank the reviewer for the question! The short answer is no. We rephrased the claim to be more accurate. Detailed explanations: We can consider a simple setting where we use FedAvg with deterministic gradients on strongly convex functions. In this case, previous literature (such as Li et al. 2020) found that we need a decayed learning rate in order to make FedAvg converge. However, when lr approaches to 0, our client consensus hypothesis will become trivial and must hold. Instead, in this paper, we showed that even with a CONSTANT learning rate, client consensus hypothesis can hold and hence, FedAvg converges fast. This result is discussed in Eqn. (13). In other words, if FedAvg converges with a constant lr, then the hypothesis must hold and FedAvg converges fast.
>
>
>     From the analysis in this paper, we can also get a more intuitive way to understand the convergence of FedAvg. The algorithm itself optimizes a surrogate loss function defined by the pseudo-gradient. This surrogate loss is smoother than the original one and can enable faster convergence. However, in order to make the optimum of this surrogate function match the original one, previous literature suggests to decay the learning rate, such that the surrogate loss gradually becomes identical to the original one at the end of training. This sacrifices the nice property of the surrogate loss. Instead, in this paper, we argue that it is possible that the surrogate loss naturally has the same optimum as the original one. So FedAvg can converge fast and there is no need to aggressively decay the learning rate.
> 5. We added more discussions around Theorem 1. Specifically, the upper bound with an optimizer learning rate was proved to match a lower bound under the same assumption (Glasgow et al. 2021). Thus, it is tight in the worst cases.
> 6. Providing a formal definition on data heterogeneity may still be a challenging research problem. We provided some discussions in the related works: data heterogeneity is implicitly captured through various assumptions in past literature (such as the gradient dissimilarity bound). To the best of our knowledge, there is no literature explicitly connecting gradient dissimilarity with underlying data heterogeneity.
>
> [1] Unreasonable effectiveness of learning neural networks: From accessible states and robust ensembles to basic algorithmic schemes
>
> [2] The Unreasonable Effectiveness of Easy Training Data for Hard Tasks
>
> [3] The Unreasonable Effectiveness of Recurrent Neural Networks

---

### Decision · Action_Editor_Z86g · 2024-05-15

**Recommendation:** Accept with minor revision

**Comment:**

This paper provides a compelling analysis of the FedAvg algorithm's performance in federated learning with heterogeneous data. It introduces the "client consensus hypothesis" and demonstrates through both theoretical analysis and empirical validation that this hypothesis can explain the observed effectiveness of FedAvg.

Summary of Reviewer Comments:

- Reviewer TjzG appreciates the clarity and simplicity of the ideas but suggests more extensive experiments and clearer examples where the hypothesis does not hold.
- Reviewer ZSqV finds the approach novel and the verification on real datasets convincing but suggests additional results to verify or falsify the hypothesis in various scenarios.
- Reviewer Vef9 considers the theory elegant and fundamental but notes that some technical interpretations are not very accurate and that the paper lacks discussion on non-convex cases.


Recommended Changes and Authors' Responses:

- The authors have added more recent references and corrected claims about SCAFFOLD's performance.
- The term "pathological datasets" was changed to "datasets with artificially high heterogeneity" to avoid controversy.
- The authors provided additional discussions to clarify the client consensus hypothesis and its applicability.
- Tables comparing existing assumptions and convergence bounds were moved to the main text for better visibility.

Conclusion:
The paper addresses a significant issue in federated learning and provides valuable insights through the proposed hypothesis and its validation. The authors have adequately responded to the reviewers' concerns, making necessary revisions to improve the clarity and comprehensiveness of the paper. Therefore, I recommend acceptance with minor revisions to further enhance the empirical validation and address the remaining technical inaccuracies.

**Audience:**

Yes.

**Claims And Evidence:**

The paper presents the "client consensus hypothesis" to explain the effectiveness of the FedAvg algorithm in federated learning settings, particularly when dealing with heterogeneous data. The hypothesis posits that the average of client updates tends to zero at the global optimum, resulting in improved convergence bounds. The authors validate this hypothesis through theoretical analysis and empirical results on both synthetic and real datasets.

Strengths:

+ The claims made in the submission are supported by clear and convincing theoretical arguments.
+ The empirical evidence provided, including experiments on linear regression, FEMNIST, and Shakespeare datasets, reinforces the theoretical findings.
+ The paper addresses a significant gap between theoretical predictions and practical observations in federated learning, providing valuable insights into the conditions under which FedAvg performs well.


Weaknesses:

- The scale of the experiments could be larger to provide a more comprehensive validation of the hypothesis across a wider range of real-world problems.
- Some technical discussions and interpretations are noted to be less accurate, and the depth of the theoretical analysis is somewhat limited.
- The paper does not propose a new algorithm but rather provides an analysis of the existing FedAvg algorithm, which might limit its perceived novelty.


Overall, the claims made in the paper are well-supported by accurate and convincing evidence, though further empirical validation on a larger scale and more diverse datasets would strengthen the conclusions.